# Conjecturing: An Overlooked Step in Formal Mathematical Reasoning

## Abstract

Autoformalisation, the task of expressing informal mathematical statements in formal language, is often viewed as a direct translation process. This, however, disregards a critical preceding step: conjecturing. Many mathematical problems cannot be formalised directly without first conjecturing a conclusion such as an explicit answer, or a specific bound. Since Large Language Models (LLMs) already struggle with autoformalisation, and the evaluation of their conjecturing ability is limited and often entangled within autoformalisation or proof, it is particularly challenging to understand its effect. To address this gap, we augment existing datasets to create ConjectureBench, and redesign the evaluation framework and metric specifically to measure the conjecturing capabilities of LLMs both as a distinct task and within the autoformalisation pipeline. Our evaluation of foundational models, including GPT-4.1 and DeepSeek-V3.1, reveals that their autoformalisation performance is substantially overestimated when the conjecture is accounted for during evaluation. However, the conjecture should not be assumed to be provided. We design an inference-time method, Lean-FIRe to improve conjecturing and autoformalisation, which, to the best of our knowledge, achieves the first successful end-to-end autoformalisation of 13 PutnamBench problems with GPT-4.1 and 7 with DeepSeek-V3.1. We demonstrate that while LLMs possess the requisite knowledge to generate accurate conjectures, improving autoformalisation performance requires treating conjecturing as an independent task, and investigating further how to correctly integrate it within autoformalisation. Finally, we provide forward-looking guidance to steer future research toward improving conjecturing, an overlooked step of formal mathematical reasoning.

## 1 Introduction

Natural language reasoning with Large Language Models (LLMs) has emerged as a powerful tool for solving complex mathematical problems. Its effectiveness is highlighted by recent breakthroughs, such as AI systems from OpenAI and Google solving five of six problems from the 2025 International Mathematics Olympiad (IMO) using natural language (Metz, 2025). The critical caveat is that these informal solutions require validation by expert mathematicians, a process that is prone to human error and lack scalability (Gouëzel & Shchur, 2019). Proof assistants like Isabelle (Wenzel et al., 2008) and Lean (Moura & Ullrich, 2021) provide a path toward automated verification at scale through formal reasoning. Their power was demonstrated when AlphaProof solved three of the six 2024 IMO problems by generating formal proofs (AlphaProof and AlphaGeometry teams, 2024) and reiterated in 2025 with SeedProver (Chen et al., 2025) equaling OpenAI and Google's performance. Yet benchmarks such as PutnamBench remain difficult, with the best open-source models achieving a correct proof rate of only 13.1% at the time of writing (Tsoukalas et al., 2024).

A central bottleneck is *autoformalisation*, the task of automatically expressing informal mathematics into a precise formal language (Szegedy, 2020). On undergraduate-level problems from the ProofNet benchmark (Azerbayev et al., 2023), the current state-of-the-art performance is only 31.28% (Liu et al., 2025b). Moreover, the fact that state-of-the-art systems like AlphaProof are provided with human-annotated formalisations, rather than the natural language problems, suggests that an end-to-end approach remains challenging. Autoformalisation is non-trivial, as even highly skilled human experts can take over eight hours to formalise a single IMO problem (Liu et al., 2025a). Improving autoformalisation would therefore be transformative, not only by providing a systematic way

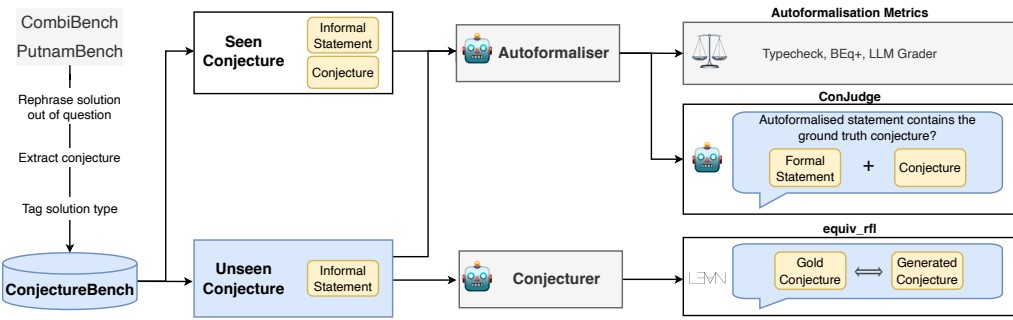

Figure 1: End-to-end evaluation pipeline for conjecturing and autoformalisation, including a "seen" setting (conjecture provided) and a more realistic "unseen" setting (conjecture must be inferred). Our contributions, highlighted in blue, introduce ConjectureBench, "unseen" evaluation, and two corresponding metrics: ConJudge for assessing conjecturing during autoformalisation and `equiv_rfl` for standalone conjecture generation.

to validate informal reasoning but also by enabling the synthesis of new data at scale to improve automated provers themselves.

Autoformalisation is difficult for two interrelated reasons: faithfulness and conjecturing. Without a ground truth formalisation[1], it can be difficult to judge whether the autoformalised statement truly reflect the intent expressed by the natural language problem (Yang et al., 2025b). Humans generally describe problems in an informal manner, often obfuscated through real world objects and situations. To formalise these, LLMs need to connect world knowledge with abstract mathematical concepts, which increases the complexity of the task (Yang et al., 2025b).

Secondly, a conjecture, a mathematical conclusion such as an explicit answer, bound, or proposition, is required for formalisation. The nature of the conjecture shapes the autoformalisation, without which proving stalls. To circumvent conjecturing during autoformalisation, one may insert a placeholder, but it must ultimately be replaced with a valid solution for a complete proof. Most current systems implicitly treat conjecturing as part of the proof search (Sun et al., 2025) by proposing a solution and validating it when a verified proof is generated. However, using a proof as self-verification of the conjecture comes with an important caveat; it does not guarantee completeness. For example, solving $x^2 - 4x = 0$ by conjecturing $x = 0$ yields a valid and verifiable yet incomplete solution, as $x = 4$ is also a valid root. This highlights that conjecturing and proving draw on distinct skills. Conjecturing relies on intuition, pattern recognition, and heuristic testing, whereas proving requires the rigorous application of tactics (Fernández-León et al., 2021).

To address the overlooked role of conjecturing in formal mathematical reasoning, we measure the conjecturing capability by introducing ConjectureBench, a new dataset designed to evaluate the conjecturing performance of LLMs. We develop two novel metrics: ConJudge, a metric that uses an LLM-as-a-Judge (Zheng et al., 2023) to assess conjecture presence within the autoformalisation, and `equiv_rfl`, a metric that uses Lean tactics to check for definitional equivalence in standalone conjecture generation as illustrated in Figure 1. Our evaluation of foundational LLMs, including GPT-4.1 and DeepSeek-V3.1, on ConjectureBench reveals that autoformalisation performance is substantially overestimated when the conjecturing step is assumed to be provided.

To test the hypothesis that this performance gap stems from a failure in reasoning rather than a lack of mathematical and world knowledge, we propose a novel inference-time method **Lean Formal-Informal Re**asoning (LEAN-FIRE). This approach guides the model by interleaving Chain-of-Thought (CoT) reasoning in natural language with Lean-of-Thought (LoT) steps in formal language, helping it to better connect informal reasoning with formal mathematics. We show that LEAN-FIRE leads to significant improvements, confirming our hypothesis. While end-to-end autoformalisation remains low, our method achieves the first successful autoformalisation of 13 new PutnamBench "no-answer" problems. More specifically, LEAN-FIRE improves conjecturing performance on our

---

[1]In this work, we always assume existence of a ground truth formalisation.

| Hypothesis | $\Rightarrow$ | Conclusion | Type of solution |
|---|---|---|---|
| $x + 4 = 0$ | $\Rightarrow$ | $x = -4$ | Numerical |
| $x^2 - a = 0$ | $\Rightarrow$ | $x \in \{\sqrt{a}, -\sqrt{a}\}$ | Algebraic |
| $\cos(x) = x$ | $\Rightarrow$ | $\exists x$ s.t. $\cos(x) = x$ | True but no closed-form solution |

Table 1: Examples of mathematical statements paired with different solution types.

ConJudge metric by an average of 29.1% for GPT-4.1 and 14.0% for DeepSeek-V3.1. These results provide strong evidence that the models' primary limitation is not a lack of requisite knowledge, but rather the need for targeted methods to unlock their ability to conjecture effectively. Lastly, through manual analysis, we further identify two practical challenges: dataset contamination and the need for new definitions, functions, and lemmata to support autoformalisation.

Our contributions are as follows: (1) we introduce ConjectureBench[2], the first benchmark evaluating conjecture capabilities, (2) we propose two complementary metrics, ConJudge and `equiv_rfl`, to systematically assess thse capabilities, and (3) we develop LEAN-FIRE, an inference-time method to improve both autoformalisation and conjecturing.

## 2 PRELIMINARY

In mathematics, a theorem is a statement for which a proof establishes a conclusion from a set of hypotheses. When such a proof is not yet known, the statement is referred to as a conjecture (Pauli, 2022). A conjecture proposes a possible conclusion often expressed as an abstract object that may or may not admit a closed-form representation such as an algebraic formula or a numerical answer, see Table 1. In formal mathematics, autoformalisation is a necessary stage prior to using a prover or proof assistant, as these systems require formal statements as inputs. Conjecturing is the task of generating candidate solutions for well-posed problems (Sun et al., 2025).

Current formal mathematics datasets largely fall into two categories. The first type assumes that a solution is already known and only requires the corresponding proof given a gold formalised statement. The second type requires the discovery of a solution before or while a proof is constructed. For this latter class, the initial step is to generate a candidate solution. Without such a conjecture, formalisation cannot proceed. This holds in Lean 4, a more permissive formal mathematics language; the compiler cannot verify whether the object types are consistent (Typecheck) in an incomplete statement.

> **Lean 4**
>
> ```
> theorem quad_roots: {x : ℝ | x^2 - 4*x = 0} ∋ conjecture := sorry
> ```

In the above `quad_roots` example, the formal statement for "Solve $x^2 - 4x$ for x", erasing `conjecture` reduces the statement to a set of hypotheses with no conclusion, leaving nothing to prove. A quick fix is to put a placeholder, `conjecture`, for which Lean 4 has been forced to assume the correct type. When the solution is known, it could be integrated directly into the formal statement. But deriving it in the first place is challenging. If generated during the proving stage, the formal language system can self-verify whether the conjecture is valid. However, the validity of a conjecture does not equate to a *complete* conjecture or a valid solution to the informal statement. Three valid and proof verifiable conjectures are:

However, only `conjecture_3` is a complete answer. In fact, natural language can frame a problem in a way that feels more intuitive and human-friendly. For example, *"How many people must be in a group for at least two of them to be born in the same month?"*, this question is easier to reason

---

[2]The dataset and code will be made available.

```
Lean 4

  abbrev conjecture_1:    abbrev conjecture_2:    abbrev conjecture_3:
    Set ℝ := {0}            Set ℝ := {4}            Set ℝ := {0, 4}
```

about using everyday knowledge than its more formal counterpart: determining the smallest domain size for which there exist no injective function into a set of 12 elements. Therefore, autoformalisation being closer to the natural language statement allows for broader possibility of generating conjectures. Finally, when tackling unsolved problems, the solution is not given in advance making conjecture generation an essential step in the formal reasoning process. Therefore, this motivates our exploration of conjecturing as an integral, yet overlooked step in formal mathematical reasoning.

## 3 METHODOLOGY

### 3.1 CONJECTUREBENCH DATASET

Two recent datasets are designed with conjecturing in mind: PutnamBench (Tsoukalas et al., 2024) factors out the solution from the problem statement, forcing models to generate the conjecture itself, while CombiBench (Liu et al., 2025a) introduces a benchmark with and without the solution to further encourage conjecture generation. To elaborate, PutnamBench is a benchmark of 640 paired informal and formal statements from the William Lowell Putnam Mathematical Competition. The benchmark and its leaderboard primarily emphasise proof generation, both when solutions are provided and when they are withheld. The evaluation of statements without answers is only feasible for 355 of the problems. Similarly, CombiBench adopts the same design where possible, with 100 combinatorics problems ranging from textbook exercises to IMO questions. However, 55 questions include the conjecture within their informal statement.

| Original with integrated solution | Reworded to seek a solution | Type of solution | Distribution |
|---|---|---|---|
| Show that there are at least 1991 red points in the plane. | What is the minimum number of red points in the plane? | Numerical | 39.0% (178) |
| Prove that there are at most $2n-1$ subsets in the collection. | What is the maximum number of subsets that can be in such a collection? | Algebraic | 36.1% (165) |
| Prove that B = $\{0, 3, 4, 9, 11\}$ is a difference set in $Z_{21}$. | Prove or disprove that B = $\{0, 3, 4, 9, 11\}$ is a difference set in $Z_{21}$. | Proof | 24.9% (114) |

Table 2: Examples of how proof questions are reformulated into the three solution types considered, along with the distribution of these types in ConjectureBench.

To adapt both datasets to evaluate conjecturing, we first annotated all 355 PutnamBench problems and 102 CombiBench problems (splitting multi-part questions into separate items) to ensure that no conjecture appear directly in the problem statements. For proof-based questions, where the conclusion is already embedded, we rephrased them into equivalent tasks requiring either a numerical or algebraic solution. When rewording is not feasible, we instead reformulate the problem into a binary classification task, requiring the model to decide whether the statement is true or false. Examples of these reformulations, as well as the distribution across our new combined dataset, ConjectureBench, are provided in Table 2. We finally separate the conjecture from the formal statement, retaining it only in the "seen" setting as illustrated in Figure 1. This design choice ensures that our full dataset of 457 paired informal–formal statements can be used consistently across both, "seen" and "unseen" settings, enabling a more accurate evaluation of conjecturing.

This evaluation framework offers several advantages. It allows us to assess whether current LLMs are capable of generating accurate conjectures while autoformalising, but also to evaluate models' raw conjecturing capability. It also enables a detailed analysis of which types of conjectures present particular challenges for existing models. The results of this benchmark provide a foundation to investigate whether improvements in conjecturing arise naturally from enhanced autoformalisation, or if alternative approaches, such as new data or reasoning approaches, are necessary.

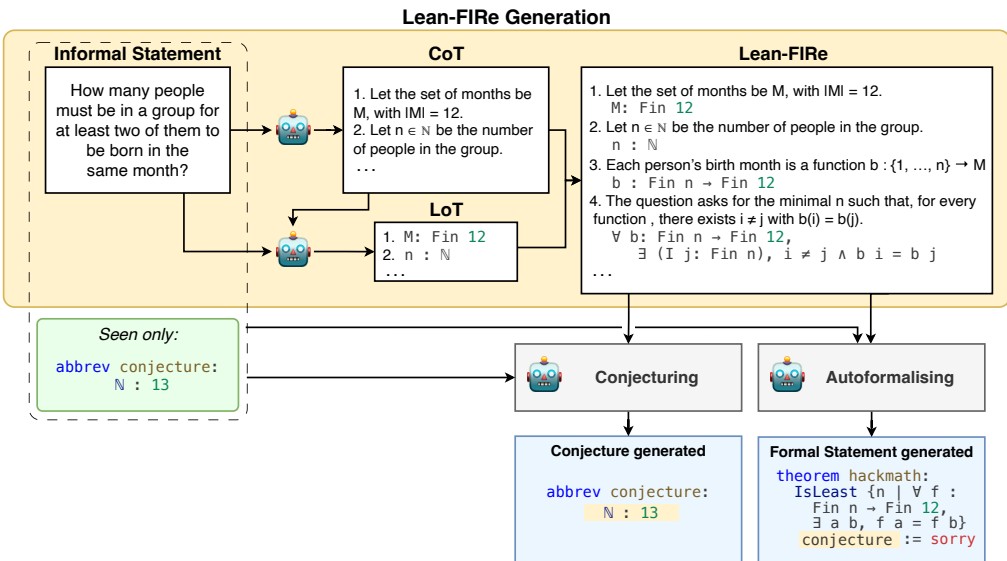

Figure 2: Illustration of LEAN-FIRE construction within the overall pipeline for generating auto-formalisations and conjectures, where the conjecture in the green box is provided only in the "seen" setting, and CoT and LoT stand for chain/lean-of-thought.

## 3.2 CONJECTURING TASKS

We evaluate performance across two distinct tasks designed to assess conjecture-driven reasoning as illustrated in Figure 2. The primary task is *autoformalisation*, which we evaluate in two settings. In the "seen" setting, the model is provided with the informal statement and the correct conjecture formatted in Lean 4. The task is to produce a formal statement that correctly incorporates the provided conjecture. In the "unseen" setting, the model is provided only with the informal statement and must deduce and incorporate the conjecture directly into the final formalisation.

The second task is *standalone conjecture generation*, where we isolate conjecturing performance entirely from the complexity of full autoformalisation. Here, the model is given only the informal statement and is instructed to generate the conjectured solution as a concise Lean 4 statement.

## 3.3 METRICS

To evaluate conjecturing performance during autoformalisation, we propose ConJudge, an LLM-as-a-judge framework (Zheng et al., 2023). Its purpose is to determine whether the problem's gold solution is reasonably and correctly incorporated as a conjecture within the final autoformalised statement. To do this, ConJudge is provided with the generated formalisation, the gold conjecture, and the gold formalisation to demonstrate the intended context and role of the conjecture. For instance, if the correct conjecture is the integer 2, the judge would reject a formalisation where 2 appears incorrectly as a power or a subscript. To tune ConJudge, we carry out a human annotation of 100 randomly sampled autoformalisation generations (Appendix B.3), classifying whether the solution was correctly incorporated into the formal statement.

For standalone conjecture generation, we created `equiv_rfl`, which evaluates definitional equivalence between the generated and gold conjecture based on tactic `rfl` (Appx. B.4). Definitional equivalence captures mathematically equivalent statements that reduce to the same value, e.g. $2 + 2 = 4$ or Nat.factorial4 $= 24$. This provides reliable formal verification through Lean's type checker. However, its limitation is that structural differences prevent equivalence verification: conjectures that are semantically identical but formatted differently are not recognized as equivalent. Human evaluation on 200 samples shows `equiv_rfl` achieves 100% precision with 71.5% recall. This provides a rigorous, formal measure of whether the model can produce the correct solution in isolation.

3.4 LEAN-GUIDED FORMAL-INFORMAL REASONING (LEAN-FIRE)

To test the hypothesis that the performance gap in conjecturing stems from a failure in reasoning rather than a lack of knowledge, we propose LEAN-FIRE, a novel inference-time method designed to better structure the model's reasoning process. The goal is to distil the LLM's latent parametric mathematical knowledge at test-time by combining both informal and formal reasoning. As illustrated in Figure 2, the LEAN-FIRE method is built as a two-stage hybrid reasoning process that integrates informal problem decomposition with formal code generation by means of interleaved Chain-of-Thought (CoT) with Lean-of-Thought (LoT) prompting. We leverage the LLM's ability in informal mathematical reasoning to first generate a potential conjecture and outline the overall structure of the formalisation. First, a complete CoT trace is generated in natural language from the informal problem statement. The CoT is designed to break down the problem, identify key mathematical objects, and articulate the reasoning entirely in natural language. Crucially, this phase is constrained to produce no formal code and avoid stating the final solution. Second, after the informal reasoning trace is completed, a subsequent LLM generates a corresponding LoT step for each informal step. The purpose of the LoT is not to write a comprehensive formal statement, but to translate the abstract concepts from the CoT into precise Lean primitives and syntax. This hybrid approach is motivated in part by prior work, such as Jiang et al. (2023), which has already demonstrated that leveraging both formal and informal language can improve performance in theorem proving.

**Seed Data Annotation.** This automated generation of CoT and LoT steps is enabled by few-shot examples derived from a small, expert-annotated seed dataset. We created this seed data from five diverse Putnam competition problems, which were annotated by an expert mathematics instructor to produce gold CoTs. The problems were selected to cover a range of mathematical domains (probability, real analysis, linear algebra, abstract algebra, number theory), solution types (as listed in Table 2), and conjecture styles, ensuring the exemplars were broadly representative. In some cases, questions were modified to omit parts of the solution, mirroring the annotation process for ConjectureBench. These five seed problems are detailed in Appendix A.1 and are excluded from our ConjectureBench evaluation. With these few-shot examples and a set of precise instructions (see Appendix A.2), CoT and LoT pairs can be automatically generated for any new problem using only its informal statement as input. In preliminary experiments, we evaluated five LLMs for this task and found that GPT-4.1 consistently outperformed its other family models and Claude-4-Opus.

## 4 EXPERIMENTAL SETUP

**Models.** We experiment with two foundational autoformalisation models: GPT-4.1 (Achiam et al., 2023) and DeepSeek-V3.1 (DeepSeek-AI et al., 2024). To measure the impact of our proposed method, we compare the performance of LEAN-FIRE against the zero-shot performance of each base model. Additionally, we conduct an ablation study where we remove the few-shot examples from the LEAN-FIRE input (w/o FS) to isolate the contribution of the hybrid reasoning approach.

**Metrics.** We assess performance for all tasks using pass@1 and pass@10, where pass@$k$ indicates that at least one of $k$ independent samples was successful. For conjecturing, we use two targeted metrics. Conjecturing performance during the full autoformalisation task is assessed with **ConJudge**, while standalone conjecture generation is evaluated using `equiv_rfl`.

For autoformalisation, we use three complementary metrics: **Typecheck**, **BEq+**, and **LLM Grader**. **Typecheck** is a binary measure of syntactic correctness indicating whether the generated Lean code compiles without error.[3] For semantic equivalence, we use **BEq+**, a metric based on a set of Lean tactics that presupposes typechecking and attempts to prove equivalence between the generated and gold formalisations (Poiroux et al., 2025). We should note that while precise, BEq+ can be overly conservative, leading to false negatives on semantically equivalent statements that differ in surface form (Liu et al., 2025b). To capture a broader notion of correctness, we also use **LLM Grader**, a pipeline that evaluates semantic alignment. First, the gold and generated formalisations are back-

---

[3]Each instance of ConjectureBench is provided with the appropriate Mathlib imports and a standardised Lean 4 environment (`v4.19.0-rc2`) to ensure consistent evaluation.

translated into natural language using a math LLM.[4] A separate judge LLM[5] then evaluates these natural language statements for semantic equivalence.

## 5 RESULTS AND DISCUSSION

### 5.1 CONJECTURING RESULTS

| Model | Method | Conjecture | ConJudge@1 | ConJudge@10 |
|---|---|---|---|---|
| GPT-4.1 | Baseline | Seen | 78.77 | 98.03 |
| | | Unseen | 26.70 $(-52.07)$ | 61.27 $(-36.76)$ |
| | LEAN-FIRE | Seen | 92.78 | 98.47 |
| | | Unseen | **55.80** $(-36.98)$ | **85.34** $(-13.13)$ |
| | LEAN-FIRE w/o FS | Seen | 77.90 | 96.06 |
| | | Unseen | 28.88 $(-49.02)$ | 56.89 $(-39.17)$ |
| DeepSeek-V3.1 | Baseline | Seen | 80.31 | 95.84 |
| | | Unseen | 30.63 $(-49.68)$ | 58.86 $(-36.98)$ |
| | LEAN-FIRE | Seen | 81.40 | 97.81 |
| | | Unseen | **44.64** $(-36.76)$ | **71.55** $(-26.26)$ |
| | LEAN-FIRE w/o FS | Seen | 74.62 | 96.72 |
| | | Unseen | 35.01 $(-39.61)$ | 56.86 $(-39.83)$ |

Table 3: Conjecturing during autoformalisation performance on ConjectureBench using ConJudge. Scores are reported at pass@1 and pass@10, with relative differences between "unseen" and "seen" in brackets. Bold indicates best performance for each model and metric in the "unseen" setting.

| Model | Type of solution | equiv_rfl@1 | equiv_rfl@10 |
|---|---|---|---|
| GPT-4.1 | All | 3.28 $(15/457)$ | 5.04 $(23/457)$ |
| | Numerical | **5.62** $(10/178)$ | **8.99** $(16/178)$ |
| | Algebraic | 3.03 $(5/165)$ | 4.24 $(7/165)$ |
| | Proof | 0.00 $(0/114)$ | 0.00 $(0/114)$ |
| DeepSeek-V3.1 | All | 3.72 $(17/457)$ | 5.70 $(26/457)$ |
| | Numerical | **7.30** $(13/178)$ | **10.67** $(19/178)$ |
| | Algebraic | 2.42 $(4/165)$ | 3.64 $(6/165)$ |
| | Proof | 0.00 $(0/114)$ | 0.88 $(1/114)$ |

Table 4: Standalone conjecture generation performance across ConjectureBench broken down by type of solution. Metrics report `equiv_rfl` at pass@1 and pass@10, with counts shown over total examples in brackets.

**Conjecturing During Autoformalisation.** Using the ConJudge metric, we find that models are more adept at producing the correct conjecture when it is part of a full autoformalisation task. Table 3 shows that LEAN-FIRE with few-shot examples significantly improves the use of conjectures in both "seen" and "unseen" settings, boosting GPT-4.1's pass@10 by up to 28% in the "unseen" setting. However, the large performance drop when few-shot examples are removed (w/o FS) indicates that the hybrid reasoning structure alone does not significantly improve conjecturing. Instead, the few-shot examples, which expose the model to various solution types and map reasoning steps to the correct conjecture format, provide the primary benefit. This suggests that a model's ability to conjecture is less a matter of latent reasoning and more a function of direct exposure, pointing to the need for larger and higher-quality conjecture datasets for training.

---

[4]We employ InternLM2-Math-Plus-20B (Cai et al., 2024).
[5]We employ a Qwen3-14B (Yang et al., 2025a) calibrated against human annotators to achieve 67.5%.

**Standalone Conjecture Generation.** As shown in Table 4, performance on standalone conjecture generation is notably low across all models. We attribute this difficulty to the lack of training data specifically for conjecturing tasks, in contrast to models' extensive exposure to autoformalisation data. This hypothesis is supported by the substantial performance improvement in the few-shot setting (Table 3), where models benefit from even minimal exposure to conjecturing examples. While models occasionally produce correct numerical conjectures, they more often generate auxiliary constructs such as definitions or lemmata instead of the conjecture itself. The performance on this task is nearly an order of magnitude lower than for conjecturing during autoformalisation (see Table 3), suggesting that models rely heavily on prior exposure to conjectures already embedded within complete formalised solutions. We observed signs of data contamination in the outputs; for instance, some generations used helper functions like `IsMagicSquare`, which appear only in the gold formalisation of the benchmark.

## 5.2 Autoformalisation Results

| Model | Method | Conjecture | TC@1 | BEq+@1 | Grader@1 | TC@10 | BEq+@10 | Grader@10 |
|---|---|---|---|---|---|---|---|---|
| **GPT-4.1** | Baseline | Seen | 25.38 | 0.00 | 7.22 | 59.52 | 6.78 | 36.32 |
| | | Unseen | 24.29$(-1.09)$ | 0.22$(+0.22)$ | 3.50$(-3.72)$ | **51.42**$(-8.10)$ | **4.38**$(-2.40)$ | 20.35$(-15.97)$ |
| | LEAN-FIRE | Seen | 31.95 | 3.72 | 11.82 | 50.98 | 6.56 | 43.33 |
| | | Unseen | 28.01$(-3.94)$ | 1.31$(-2.41)$ | 4.60$(-7.22)$ | 43.76$(-7.22)$ | 3.06$(-3.50)$ | 22.76$(-20.57)$ |
| | LEAN-FIRE w/o FS | Seen | 35.89 | 2.84 | 7.66 | 49.02 | 4.60 | 40.04 |
| | | Unseen | **28.45**$(-7.44)$ | **2.41**$(-0.43)$ | **5.69**$(-1.97)$ | 42.67$(-6.35)$ | 4.16$(-0.44)$ | **23.85**$(-16.19)$ |
| **DeepSeek-V3.1** | Baseline | Seen | 38.29 | 4.81 | 6.78 | 61.71 | 6.78 | 35.67 |
| | | Unseen | 33.26$(-5.03)$ | 2.63$(-2.18)$ | 5.25$(-1.53)$ | 54.49$(-7.22)$ | **5.47**$(-1.31)$ | 24.95$(-10.72)$ |
| | LEAN-FIRE | Seen | 46.17 | 3.72 | 9.85 | 66.74 | 6.13 | 41.36 |
| | | Unseen | **42.89**$(-3.28)$ | 2.63$(-1.09)$ | 6.13$(-3.72)$ | **59.30**$(-7.44)$ | 4.16$(-1.97)$ | **26.91**$(-14.45)$ |
| | LEAN-FIRE w/o FS | Seen | 39.82 | 3.50 | 9.41 | 56.24 | 4.16 | 39.39 |
| | | Unseen | 39.61$(-0.21)$ | 2.63$(-0.87)$ | **6.35**$(-3.06)$ | 53.83$(-2.41)$ | 3.72$(-0.44)$ | 23.63$(-15.76)$ |

Table 5: Autoformalisation performance of all models and methods (as percentages) on ConjectureBench across seen and unseen settings. Metrics include TC (Typecheck), BEq+, and Grader (LLM Grader), reported at pass@1 and pass@10. Unseen results show the difference relative to seen performance in brackets. Bold values indicate the best performance for each model and metric in the "unseen" setting.

Table 5 shows that correct end-to-end autoformalisation remains a challenging task, with low success rates even in the "seen" setting where the conjecture is provided. Performance is systematically overestimated in this setting, with an average 23.7% drop in performance when moving from the "seen" to the "unseen" setting. Despite these challenges, LEAN-FIRE achieves notable successes. Generating conjectures, as underscored by the PutnamBench "no-answer" leaderboard, was considered as a challenge with no successful submissions to date (Tsoukalas et al., 2024). Yet, even under the strict BEq+ metric, LEAN-FIRE enables GPT-4.1 to correctly autoformalise 13 new PutnamBench problems and DeepSeek-V3.1 to solve 7. To our knowledge, these represent the first successful autoformalisations on PutnamBench in a setting where the solution is withheld.

In contrast to its effect on conjecturing, LEAN-FIRE's impact on autoformalisation is more nuanced. When comparing across metrics, both models show consistent gains under Typecheck and LLM Grader. Higher Typecheck scores indicate improved syntactic correctness, while better LLM Grader scores point to improved semantic equivalence. Therefore, the limited gains in BEq+ suggest that assembling correct components into a fully equivalent formalisation remains a key bottleneck. For example, in the generated formalisation of `putnam_2014_b2` below, both Typecheck and LLM Grader marked the output as correct, but BEq+ did not due to a subtle error: a misplaced factorial symbol. This highlights the sensitivity of BEq+ and illustrates that even when all components are present, models may fail to assemble them with complete accuracy.

```lean
Lean 4

abbrev conjecture: (fun n : ℕ => (-1)^(n - 1) / ((n - 1)! * n!))

theorem putnam_2014_a2 : ∀ n : ℕ, 0 < n
    → let A : Matrix (Fin n) (Fin n) ℚ := λ i j
    => 1 / (min (i.val + 1) (j.val + 1) : ℚ) in det A
    = ((-1) ^ (n - 1) : ℚ) / ((n - 1)! * n)!)
    := sorry
```

In general, the comparison with the baseline reveals no consistent performance benefit. In the "seen" setting, few-shot examples are helpful, but in the "unseen" setting, they can be detrimental, sometimes wrongly encouraging template solutions where a conjecture is introduced as a separate function and then integrated into the formalisation. This suggests that the mathematical knowledge required for complex autoformalisation including conjecturing is not fully latent in the model's parameters, or that LEAN-FIRE, in its current form, fails to consistently extract it. LEAN-FIRE shows a net mean gain of 3.01% at pass@1 but a slight decline at pass@10, suggesting that the reasoning guidance primarily helps steer the model's token distribution towards correctness, but the effect is diluted when multiple generations are sampled by the increase of the probability of reaching a better distribution. Still, from Table 5, best-of-n sampling roughly doubles improvement under BEq+ and quadruples it under the LLM Grader, indicating that necessary knowledge exists in latent space, but is hard to reliably retrieve.

## 6 RELATED WORK

Several approaches to autoformalisation leverage retrieval or supervised fine-tuning to bootstrap formal reasoning. For example, Liu et al. (2025b) incorporate retrieval to ground the translation process, while Lin et al. (2025) train on large corpora containing both human and synthetic annotations derived from the Lean Workbook (Ying et al., 2024), exposing the model to a diverse range of formalisation examples. Data-centric strategies, focusing on increasing dataset size or improving data quality, are also common. Some methods employ LLMs-as-a-judge (Wang et al., 2025), chain-of-thought (CoT) model scoring (Xin et al., 2024), Lean typechecking signals (Lu et al., 2024), or LLM feedback (Peng et al., 2025). In addition, Sun et al. (2025) combine typechecking feedback with retrieval within their framework to further enhance autoformalisation performance.

Autoformalisation is also employed in theorem proving: for instance, Jiang et al. (2023) propose a "draft–sketch–prove" framework that first sketches proof outlines from informal arguments before completing subgoals with an automated prover. Collectively, these works highlight a growing toolkit of data generation, model training, and feedback mechanisms aimed at closing the gap in autoformalisation. However, these work fail to improve models using test-time compute which we tackle with LEAN-FIRE.

Conjecturing in the broader sense has been aimed to formalise open-ended conjectures to encourage mathematical discovery (Chau et al., 2025). Methodologically, many approaches interleave conjecturing with proving, where a placeholder conjecture is proposed and subsequently validated by a prover (Dong & Ma, 2025). Sun et al. (2025) extend this idea by iteratively generating special coded cases from an autoformalised statement, forming candidate conjectures that are then tested by a prover in a repeated cycle. While these works incorporate conjecturing as part of their pipelines, they do not isolate or systematically evaluate the conjecturing step itself. Zhou et al. (2024) demonstrate that for simple enough problems, LLMs could be used to generate the solutions and autoformalisation can verify them. Our work is the first to explicitly extract solution conjecturing as a distinct capability, provide dedicated evaluation metrics, and systematically benchmark model performance.

## 7 CONCLUSION

In this work, we identify conjecturing as an overlooked step in formal mathematical reasoning with LLMs, challenging the prevailing assumption that autoformalisation is a straightforward translation task. By introducing ConjectureBench, a benchmark specifically designed to evaluate conjecture generation, and by proposing new metrics that disentangle conjecturing from autoformalisation, we provide the first systematic framework to measure and analyse this capability. Our results show that existing models substantially underperform when conjectures are withheld, revealing that much of their perceived success depends on having solutions pre-specified. To address this gap, we develop LEAN-FIRE, an inference-time strategy that integrates informal Chain-of-Thought with formal Lean-of-Thought reasoning. This method enables the first successful end-to-end autoformalisation of PutnamBench "no-answer" problems, demonstrating that LLMs possess latent mathematical knowledge but require structured guidance to effectively conjecture and formalise. Manual analysis also identify two challenges: data contamination of existing benchmarks, and the task of generating useful definitions, functions and lemmata that would help autoformalisation, conjecturing and proving. For future work, we argue that progress in formal mathematical reasoning hinges on treating conjecturing as an independent task. This calls for the development of richer conjecturing datasets, improved inference-time techniques, and training strategies that explicitly separate and then reintegrate conjecturing with autoformalisation.

### ETHICS STATEMENT

In conducting this research, we strictly adhere to data protection regulations in the respective countries and follow established academic codes of ethics. We respect the licenses of all data artifacts utilised ensuring that their usage complies with the terms set by the creators. LLMs were solely used to assist in editing and improving the language of this manuscript. All experts involved in data annotation and validation were fairly compensated for their contributions.

While we acknowledge that reasoning-oriented LLMs can potentially be misused to generate harmful content, we believe that the associated risks are minimal in the context of improving formal mathematical reasoning capabilities. Compared to related works in this area, we do not identify any additional ethical risks arising from our models, datasets, or methodologies.

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

## A  LEAN-FIRE

In this Appendix section, we provide the five examples used both as seed questions and few-shot examples A.1. We also include the prompts used to generate the CoT and the subsequent LoT in A.2.

### A.1  SEED QUESTIONS

**Seed/Few-shot example 1 of 5**

**Name**

putnam_2004_a1

**Informal Statement**

Basketball star Shanille O'Keal's team statistician keeps track of the number, $S(N)$, of successful free throws she has made in her first $N$ attempts of the season. Early in the season, $S(N)$ was less than 80% of $N$, but by the end of the season, $S(N)$ was more than 80% of $N$. Proof or disprove that it there necessarily was a moment in between when $S(N)$ was exactly 80% of $N$.

**LeanFIRe Reasoning**

```
- Each attempt has a value in {0,1}, 0 for fail, 1 for success, i.e.
  attempt: ℕ → {0,1}.
  Lean: attempt : ℕ → Fin 2

- The function S is the average score of the attempt, i.e., the sum
  of the attempts divided by the number of attempts S: attempts → ℝ.
  Lean: S : (ℕ → Fin 2) → ℕ → ℝ
    S attempts N = (∑ {i : Fin N} (attempts i).1) / N

- S(N) can be written as S(N) = m_N / N where m_N is the number of
  successes in N tries, i.e. m_N = ∑_{i=1}^{N} 1_{success}.
  Lean: (encoded in definition of S above)

- The success rate is below 80% at one point a, S(a) < 0.8, and
  above 80% at another point b > a, S(b) > 0.8.
  Lean: 1 ≤ a ∧ a < b ∧ S attempts a < 0.8 ∧ S attempts b > 0.8

- Show there exists c ∈ (a,b) with S(c) = 0.8.
  Lean: ∃ c : ℕ, a < c ∧ c < b ∧ S attempts c = 0.8
```

**Conjecture**

```
abbrev conjecture : Prop := True
```

**Formal Statement**

```
theorem putnam_2004_a1
  (S : (ℕ → Fin 2) → ℕ → ℝ)
  (hS : ∀ attempts, ∀ N ≥ 1, S attempts N = (Σ i : Fin N, (attempts
   i).1) / N) :
  (∀ attempts a b,
    (1 ≤ a ∧ a < b ∧ S attempts a < 0.8 ∧ S attempts b > 0.8) →
      (∃ c : ℕ, a < c ∧ c < b ∧ S attempts c = 0.8))
  ↔ conjecture :=
sorry
```

Figure 3: Example (1/5) from Putnam annotated with informal and formal hint reasoning steps.

**Seed/Few-shot example 2 of 5**

**Name**

putnam_2009_b2

**Informal Statement**

A game involves jumping to the right on the real number line. If $a$ and $b$ are real numbers and $b > a$, the cost of jumping from $a$ to $b$ is $b^3 - ab^2$. For what real numbers $c$ can one travel from 0 to 1 in a finite number of jumps with total cost exactly $c$?

**LeanFIRe Reasoning**

– The jumps can be modelled as a sequence that partitions the interval $(0,1)$, with $N \in \mathbb{N}$ jumps, $s_0 = 0$, $s_i = 1$, and $s_i < s_{i+1}$ for all $0 \le i < N$.
  Lean: s : Fin (N + 1) → ℝ
    validPath (s : Fin (N + 1) → ℝ) : Prop :=
    s 0 = 0 ∧ s (Fin.last N) = 1 ∧ ∀ i : Fin N, s i < s (i.succ)

– The cost of a jump from $s_i$ to $s_{i+1}$ is $s_{i+1}^3 - s_i * s_{i+1}^2$.
  Lean: jumpCost (a b : ℝ) : ℝ := b^3 - a * b^2

– The total cost for all jumps is $\sum_{i=0}^{N-1}(s_{i+1}^3 - s_i * s_{i+1}^2)$.
  Lean: totalCost (s : Fin (N + 1) → ℝ) : ℝ :=
    ∑ {i : Fin N} jumpCost (s i) (s (i.succ))

– The set of reachable costs is { c ∈ ℝ | ∃ N ∈ ℕ, validPath s ∧ totalCost(s) = c }.
  Lean: reachableCosts : Set ℝ :=
    {c : ℝ | ∃ (N : ℕ) (s : Fin (N + 1) → ℝ),
    validPath s ∧ totalCost s = c}

**Conjecture**

```
abbrev conjecture : Set ℝ := Ioc (1 / 3) 1
```

**Formal Statement**

```
theorem putnam_2009_b2
: ({c : ℝ | ∃ s : ℕ → ℝ, s 0 = 0 ∧ StrictMono s ∧ (∃ n : ℕ, s n =
    1 ∧ ((∑ i ∈ Finset.range n, ((s (i + 1)) ^ 3 - (s i) * (s (i +
    1)) ^ 2)) = c))} = conjecture) :=
sorry
```

Figure 4: Example (2/5) from Putnam annotated with informal and formal hint reasoning steps.

**Seed/Few-shot example 3 of 5**

**Name**

putnam_2013_b2

**Informal Statement**

Let $C = \bigcup_{N=1}^{\infty} C_N$, where $C_N$ denotes the set of those 'cosine polynomials' of the form

$$f(x) = 1 + \sum_{n=1}^{N} a_n \cos(2\pi nx)$$

for which:

(i) $f(x) \geq 0$ for all real $x$, and

(ii) $a_n = 0$ whenever $n$ is a multiple of 3.

Determine the maximum value of $f(0)$ as $f$ ranges through $C$, and prove that this maximum is attained.

**LeanFIRe Reasoning**

```
- C is the set of all C_N for a given N ∈ ℕ.
  Lean: C_N (N : ℕ) : Set (ℝ → ℝ) :=
    { f | ∃ (a : ℕ → ℝ),
      (∀ x, f x = 1 + ∑{n ∈ Finset.range N} a n * Real.cos (2 * π *
      n * x)) ∧
      (∀ x, f x ≥ 0) ∧ (∀ n, n % 3 = 0 → a n = 0) }

- C_N is defined as the set of polynomials of the form f(x) = 1 +
  ∑_{n=1}^{N} a_n cos(2πnx) where f(x) ≥ 0 for all x ∈ ℝ, and the coefficient
  a_n = 0 whenever n is a multiple of 3.
  Lean: (above definition of C_N already encodes this)

- Therefore, C_N = { f(x) ∈ ℝ | f(x) = 1 + ∑_{n=1}^{N} a_n cos(2πnx),  f(x) ≥ 0
  ∀ x ∈ ℝ,  a_n = 0 if n mod 3 = 0 }.
  Lean: (same C_N definition)

- C is the union of all the C_N, i.e. C = ⋃_{N=1}^{∞} C_N.
  Lean: C : Set (ℝ → ℝ) := ⋃ N, C_N N

- Determine the maximum f(0) within all possible C_N, i.e. sup {
  f(0) | f ∈ C }.
  Lean: supF0 : ℝ := Sup { f 0 | f ∈ C }
```

**Conjecture**

```
abbrev conjecture : ℝ := 3
```

**Formal Statement**

```
theorem putnam_2013_b2
  (CN : ℕ → Set (ℝ → ℝ))
  (hCN : ∀ N : ℕ, CN N =
    {f : ℝ → ℝ |
      (∀ x : ℝ, f x ≥ 0) ∧
      ∃ a : List ℝ, a.length = N + 1 ∧ (∀ n : Fin (N + 1), 3 | (n :
    ℕ) → a[n]! = 0) ∧
      ∀ x : ℝ, f x = 1 + ∑ n ∈ Finset.Icc 1 N, a[(n : ℕ)]! *
    Real.cos (2*Real.pi*n*x)}) :
  IsGreatest {f 0 | f ∈ ⋃ N ∈ Ici 1, CN N} conjecture :=
sorry
```

Figure 5: Example (3/5) from Putnam annotated with informal and formal hint reasoning steps.

**Seed/Few-shot example 4 of 5**

**Name**
putnam_2014_a2

**Informal Statement**
Let $A$ be the $n \times n$ matrix whose entry in the $i$-th row and $j$-th column is $\frac{1}{\min(i,j)}$ for $1 \leq i, j \leq n$.
Compute $\det(A)$.

**LeanFIRe Reasoning**

– Let the dimension of the matrix be $n \in \mathbb{N}$, and the $n \times n$ matrix
  $A \in \mathbb{R}^{n \times n}$.
  Lean: A (n : ℕ) : Matrix (Fin n) (Fin n) ℝ :=

– Define $A_{ij}$ to be the entry from the i-th row and j-th column of
  matrix A.
  Lean: (implicit in the matrix function arguments λ i j)

– Set each entry to be the minimum between its column and row
  value, i.e. $A_{ij}$ = 1 / $\min(i,j)$ ∀ 1 ≤ i, j ≤ n.
  Lean: λ i j => 1 / min (i.1 + 1) (j.1 + 1)
  Note: i.1 + 1 and j.1 + 1 are used because Lean indices start at
  0 but min(i,j) starts at 1

– Evaluate $\det(A)$.
  Lean: detA (n : ℕ) : ℝ := Matrix.det (A n)

**Conjecture**

```
abbrev conjecture : ℝ := 3
```

**Formal Statement**

```
theorem putnam_2013_b2
  (CN : ℕ → Set (ℝ → ℝ))
  (hCN : ∀ N : ℕ, CN N =
    {f : ℝ → ℝ |
      (∀ x : ℝ, f x ≥ 0) ∧
      ∃ a : List ℝ, a.length = N + 1 ∧ (∀ n : Fin (N + 1), 3 | (n :
    ℕ) → a[n]! = 0) ∧
      ∀ x : ℝ, f x = 1 + Σ n ∈ Finset.Icc 1 N, a[(n : ℕ)]! *
    Real.cos (2*Real.pi*n*x)}) :
  IsGreatest {f 0 | f ∈ ∪ N ∈ Ici 1, CN N} conjecture :=
sorry
```

Figure 6: Example (4/5) from Putnam annotated with informal and formal hint reasoning steps.

**Seed/Few-shot example 5 of 5**

**Name**

putnam_2015_a2

**Informal Statement**

Let $a_0 = 1$, $a_1 = 2$, and $a_n = 4a_{n-1} - a_{n-2}$ for $n \geq 2$. Find an odd prime factor of $a_{2015}$.

**LeanFIRe Reasoning**

```
– A recurrence relation is initialised with 1 and 2 as the starting
  points, i.e. a₀ = 1 and a₁ = 2.
  Lean: a : ℕ → ℕ
    a 0 = 1
    a 1 = 2

– It is defined as 4 times the previous term minus the term before
  the previous one, i.e. aₙ = 4aₙ₋₁ − aₙ₋₂ for n ≥ 2.
  Lean: ∀n ≥ 2, a n = 4 ∗ a (n − 1) − a (n − 2)

– For the 2015th term of the sequence, a₂₀₁₅, determine a factor c ∈ ℕ
  such that:
  • c | a₂₀₁₅
  • c is odd (∃ n ∈ ℕ, c = 2n − 1)
  • c is prime (no divisor k > 1 except itself)
  Lean: ∃ p : ℕ, p | a 2015 ∧ Nat.Prime p ∧ Odd p
```

**Conjecture**

```
abbrev conjecture : ℕ := 181
```

**Formal Statement**

```
theorem putnam_2015_a2
(a : ℕ → ℤ)
(abase : a 0 = 1 ∧ a 1 = 2)
(arec : ∀ n ≥ 2, a n = 4 ∗ a (n − 1) − a (n − 2))
: Odd conjecture ∧ conjecture.Prime ∧ ((conjecture : ℤ) | a 2015) :=
sorry
```

Figure 7: Example (5/5) from Putnam annotated with informal and formal hint reasoning steps.

## A.2 LEAN-FIRE PROMPTS

---

### Chain-of-Thought (CoT) Generation Prompt

**System Prompt**

---

```
You are an advanced assistant specializing in formal mathematics and Lean 4
theorem proving. You have extensive expertise in translating mathematical
concepts from natural language into precise Lean 4 code.
```

---

**User Prompt**

```
Using the provided informal statement, write a concise sequence of hints that
guides the reader towards a formal statement in Lean.
Guidelines:
Do not include any Lean code.
Hints must be succinct and make use of mathematical notation.
Do not include proof steps|ignore any part that concerns only the proof.
Ensure that all variables, functions, and assumptions are clearly introduced
and well-defined.
Use the hints to bridge the gap between the worded (informal) problem and
the underlying mathematics|make clear how each mathematical concept
corresponds to elements of the informal statement.
Refer to the following examples of previously generated hints for style
and structure.
{%- for example in examples %}
EXAMPLE {{ example.id }}:
**Informal statement**
{{ example.informal_statement }}
**Hints**
{{ example.cot}}
{%- endfor }
**Informal statement**
{{ query.informal_statement }}
**Hints**
```

---

Figure 8: Jinja templates for the system and user prompt used in LeanFIRE for the generation of informal reasoning steps (CoT).

---

**Lean-of-Thought (LoT) Translation Prompt**

**System Prompt**

```
You are an advanced assistant specializing in formal mathematics and Lean 4
theorem proving. You have extensive expertise in translating mathematical
concepts from natural language into precise Lean 4 code.
```

---

**User Prompt**

```
Using the provided hints, write a Lean4 code snippets for each hints when
appropriate to guide the reader towards a formal statement in Lean.
Guidelines:
Do not provide formal proofs or imports.
Ensure that you match the hints to the Lean hints.
Refer to the following examples of previously generated hints for style
and structure.
{%- for example in examples %}
EXAMPLE {{ example.id }}:
**Informal statement**
{{ example.informal_statement }}
**Hints**
{{ example.cot}}
**Lean Hints**
{{ example.lot}}
{%- endfor }
**Informal statement**
{{ query.informal_statement }}
**Hints**
{{ example.cot}}
**Lean Hints**
```

---

Figure 9: Jinja templates for the system and user prompt used in LeanFIRE for the translation of the CoT into formal reasoning steps (LoT).

## B    DETAILS ON EXPERIMENTAL SETUP

This Appendix provides details on our experimental setup. All experiments were conducted in Lean v4.19.0-rc2 with the appropriate Mathlib imports and standard LLM APIs for GPT-4.1 and DeepSeek-V3.1. Each instance was run for 10 passes using the random seeds [5049, 891, 1065, 4894, 3277, 8476, 8192, 688, 377, 3568] to ensure reproducibility. The only non-default generation parameter was a temperature of 0.7; all other settings were kept at their default values. Prompts for autoformalisation, conjecture generation, and ConJudge are provided in Sections B.1, B.2, and B.3, respectively. The Lean 4 code for equiv_rfl is included in Section B.4.

## B.1 AUTOFORMALISATION PROMPT

---

**Autoformalisation Prompt**

**System Prompt**

You are an advanced assistant specializing in formal mathematics and Lean 4
theorem proving. You have extensive expertise in translating mathematical
concepts from natural language into precise Lean 4 code.

---

**User Prompt**

```
Translate the following natural language statement, provided under
**Informal statement** into a formal Lean 4 theorem. Use the theorem name
specified under **Name** as the Lean identifier for the theorem. Your
response must:
- Write only valid Lean 4 code, with clear and idiomatic use of Lean
syntax and conventions.
- Only include the formalization, and do not include any proof or imports.
- Define the theorem using the provided name.
- Faithfully capture the meaning of the informal statement in your
formalization.
- Enclose all Lean code within triple backticks
Output:
```lean
theorem [NAME] : [Lean formalization of the statement] := sorry
```
{%- for example in examples %}
EXAMPLE {{ example.id }}:
**Name**
{{ example.name }}
**Informal statement**
{{ example.informal_statement }}
The code below presents a solution implementation written in Lean 4.
This solution has already been incorporated into the current Lean
environment and is available for use in the formalization.
import Mathlib
{%- if conjecture_is_seen %}
{{ example.conjecture }}
{%- endif %}
Output:
```lean
{{ example.formal_statement }}
```
Above are examples for you to model the translation of the below natural
language statement into a Lean 4 formal theorem:
{%- endfor }
**Name**
{{ query.name }}
**Informal statement**
{{ query.informal_statement }}
The code below presents a solution implementation written in Lean 4.
This solution has already been incorporated into the current Lean
environment and is available for use in the formalization.
import Mathlib
{%- if conjecture_is_seen %}
{{ example.conjecture }}
{%- endif %}
**Combined Hints**
{{ query.combined_cot_lot }}
Output:
```lean
```

---

Figure 10: Jinja templates for the system and user prompt for autoformalisation.

## B.2 Standalone Conjecture Generation Prompt

---

**Conjecturing Prompt**

**System Prompt**

```
You are an advanced assistant specializing in formal mathematics and Lean 4
theorem proving. You have extensive expertise in translating mathematical
concepts from natural language into precise Lean 4 code.
You do not provide proofs or full theorem statements, only the mathematical
expression representing the solution, proposition, or the value being asserted.
You should first analyze the informal problem statement, then provide the final
expression as valid Lean 4 code.
```

---

**User Prompt**

```
Your task is to take a natural language mathematical statement and extract the
mathematical expression, proposition, or value, representing it as a Lean 4
expression.
**Instructions:**
1. Analyze the informal problem statement to deconstruct its mathematical components.
2. Provide the final solution as a single Lean 4 expression.
3. Present the final output inside a Lean code block, using:
```lean
abbrev solution {solution code}
```
**Informal statement**
{{ example.informal_statement }}
```

---

Figure 11: Jinja template for the system and user prompt used in to generate a conjecture in Lean 4.

## B.3 ConJudge

ConJudge evaluates whether a conjecture appears in a given formalised statement. We first conducted human annotations to identify which model and prompt best align with human judgments; this model was then selected as our LLM-as-a-judge. Table 6 presents the distribution of human annotations for 100 sample generations, while Table 7 reports the accuracy of four different models against the human gold labels. The prompt used for ConJudge is provided below.

|        | TRUE | FALSE | Total |
|--------|------|-------|-------|
| **Seen**   | 35 | 11 | 46 |
| **Unseen** | 21 | 33 | 54 |
| **Total**  | 56 | 44 | 100 |

Table 6: Contingency table showing counts of TRUE and FALSE values for seen and unseen instances.

| Model | Percentage |
|-------|------------|
| internlm2-math-plus-20b | 60 |
| qwen3-14b | 79 |
| gpt-oss-20b | 70 |
| qwen3-30b-a3b-instruct | **83** |

Table 7: Percentage alignment to human annotators for ConjectureBench across different models.

---

**ConJudge Evaluation Prompt**

**System Prompt**

```
You are an expert in the Lean 4 theorem proving language and formal
mathematics. Your task is to determine if a given formal statement in
Lean 4 contains a specific conjectured value, algebraic formula, or bound.
You will be given three inputs:
1. **Conjecture**: The value, formula, or bound to look for.
2. **Ground Truth Formal Statement**: An example of a Lean 4 statement that
correctly formalizes the conjecture. Use this as a reference for a valid
implementation.
3. **Formal Statement**: The Lean 4 code you need to evaluate.
Your goal is to determine if the **Formal Statement** contains the core
assertion of the **Conjecture**. The **Ground Truth Formal Statement** is
provided to help you understand how the conjecture can be formally expressed.
The statement you are evaluating might not have the exact same syntax as the
ground truth. You must carefully check for **semantically equivalent
variations** of the conjecture's core idea. This includes, but is not limited
to, permutations of terms, different but equivalent algebraic expressions, or
reordered hypotheses. Additionally, a conjecture can be expressed either by
defining a proposition (e.g., 'abbrev conjecture : Prop := ...') or by
asserting it within a theorem, which implicitly states the conjecture holds.
You should consider these forms equivalent.
Your output must follow this structure exactly:
1. First, provide a brief explanation of your reasoning.
2. Second, conclude with the final answer in the format: 'The formal
statement contains the conjecture: **True**' or 'The formal statement
contains the conjecture: **False**'.
```

---

**User Prompt**

```
**Conjecture:**
```lean
{{ conjecture }}
```
**Ground Truth Formal Statement:**
```lean
{{ statement1 }}
```
**Formal Statement:**
```lean
{{ statement2 }}
```
```

---

Figure 12: Jinja templates for the system and user prompts used by CONJUDGE.

## B.4 EQUIV_RFL

> **Lean 4**
>
> ```
> abbrev conjecture_gold: {gold}
> abbrev conjecture_generated: {generated}
>
> theorem thm : conjecture_gold = conjecture_generated := by rfl
> ```

Figure 13: Implementation of metric equiv_rfl in Lean 4.

