# OpenReview forum: "Conjecturing: An Overlooked Step in Formal Mathematical Reasoning"
_ICLR.cc/2026/Conference — Submitted to ICLR 2026_

### Official Review · Reviewer_BRQu · 2025-10-29

**Soundness:** 2
**Presentation:** 1
**Contribution:** 2
**Rating:** 2
**Confidence:** 4

**Summary:**

This paper tackles the problem of autoformalization by being the first to explicitly identify and decouple conjecturing as a critical, overlooked step. It defines conjecturing as the process of generating a concrete solution before formalization. The authors introduce a new benchmark, ConjectureBench, and two metrics (ConJudge, equiv rfl) to measure this specific capability. Finally, they propose LEAN-FIRE, a few-shot, inference-time reasoning method combing informal and formal chains of thought, which demonstrates improved performance on some no-answer problems in PutnamBench compared to a baseline that directly calls the LLM.

**Strengths:**

The primary strength of this paper is its novel formulation of the problem. Identifying conjecturing as a separate bottleneck from formalization is a key insight that is valuable to the community and the experiments are effective. This conceptual reframing, supported by the intuitive "seen" vs. "unseen" experimental design, provides a novel lens for analyzing failures in end-to-end reasoning.

**Weaknesses:**

1. The authors' own ablation study (Table 3 and Table 5, 'Baseline' vs. 'LEAN-FIRE w/o FS') shows that the hybrid reasoning structure ('w/o FS') provides almost no benefit, and the gains come entirely from the few-shot examples, which suggests LEAN-FIRE is not an effective reasoning framework, but rather a form of highly-tuned prompt engineering. This finding undermines the core claim that the framework can "distill the LLM's latent parametric mathematical knowledge".
2. The paper’s empirical claims rely on evaluation metrics with major methodological issues, undermining the validity of its conclusions. Specifically, ConJudge shows clear inaccuracy, while BEq+ is overly strict and prone to false negatives. In addition, the LLM Grader’s back-translation approach has not been empirically verified. Additionally, The equiv_rfl task is claimed to assess the model’s conjecturing ability independently of autoformalization, yet the methodology still appears to involve a rigid formal-language check, which blurs this distinction. Furthermore, the conjectures illustrated in Figure 2 seem easily verifiable through informal LLM reasoning. As the authors themselves note on Page 7, “(models) more often generate auxiliary constructs such as definitions or lemmata instead of the conjecture itself,” suggesting that the reported performance on this task may not reliably reflect the models’ true conjecturing ability.
3. The paper's methodological contribution to the conjecturing step (pre-proof solution-finding) appears limited. The paper frames this step as a key bottleneck, but the proposed solution does not seem to offer a novel or structured technique for solving it. Instead, the methodology appears to rely on the Large Language Model's (LLM's) inherent, general-purpose reasoning capabilities to inference these closed-form problems. If the core of the conjecturing method is, in essence, to prompt the LLM for the solution, then the contribution is less of a new conjecturing scheme and more an application of an existing LLM.
4. The integrity of the ConjectureBench dataset is questionable due to its construction via manual "rephrasing" of existing problems. This method is a weak defense against data contamination, which is a problem the authors themselves admit to observing in their results. It cannot guarantee that the "unseen" setting is truly new to the models. Consequently, the reported performance likely underestimates the actual difficulty of the conjecturing task.

**Questions:**

1. The equiv_rfl scores in Table 4 are notably low. Could you elaborate on the reason for this? Specifically, could you elaborate on your analysis of potential false negatives where logically correct conjectures were rejected due to syntactic differences from the gold standard?
2. The proposed metrics present clear trade-offs: ConJudge has a notable disagreement rate with human annotators, while equiv_rfl is strict. How do you propose combining these complementary signals into a single, reliable evaluation? For instance, how could the formal certainty from equiv_rfl be used to calibrate the potential inaccuracies in ConJudge's semantic judgments?

---

> ### Author Response · Authors · 2025-11-23
>
> 1. The authors' own ablation study (Table 3 and Table 5, 'Baseline' vs. 'LEAN-FIRE w/o FS') shows that the hybrid reasoning structure ('w/o FS') provides almost no benefit, and the gains come entirely from the few-shot examples, which suggests LEAN-FIRE is not an effective reasoning framework, but rather a form of highly-tuned prompt engineering. This finding undermines the core claim that the framework can "distill the LLM's latent parametric mathematical knowledge".
>
> We believe that Lean-FIRe does add value by directing the model to the correct aspect of its parametric knowledge to solve the problems and autoformalise them, however when shown what the task involves via the few-shot it's better. In our ablation, we acknowledge that there are cases where the FS are harmful but from the results this is not the norm. For instance, from Table 5, Lean-FIRe's performance for unseen is only lower than the baseline 3 times out of 12 where two of them are for BEq+ which is a very strict metric.
>
> 2. The paper’s empirical claims rely on evaluation metrics with major methodological issues, undermining the validity of its conclusions. Specifically, ConJudge shows clear inaccuracy, while BEq+ is overly strict and prone to false negatives. In addition, the LLM Grader’s back-translation approach has not been empirically verified. Additionally, The equiv_rfl task is claimed to assess the model’s conjecturing ability independently of autoformalization, yet the methodology still appears to involve a rigid formal-language check, which blurs this distinction. Furthermore, the conjectures illustrated in Figure 2 seem easily verifiable through informal LLM reasoning. As the authors themselves note on Page 7, “(models) more often generate auxiliary constructs such as definitions or lemmata instead of the conjecture itself,” suggesting that the reported performance on this task may not reliably reflect the models’ true conjecturing ability.
>
> For autoformalisation, we use well-established metrics i.e Typecheck, Beq+ (Rethinking and Improving Autoformalization: Towards a Faithful Metric and a Dependency Retrieval-based Approach:https://openreview.net/pdf?id=hUb2At2DsQ) and LLM Grader (FormalMATH:https://arxiv.org/pdf/2505.02735). BEq+ has a 100% precision so it's very strict and even though it is prone to false negatives, it provides a very solid lower bound. The LLM Grader approach was first established by FormalMATH (named semantic verification) which already has 20 citations according to google scholar. However, despite stating that in footnote 5, line 377 that we calibrated the LLM grader against human annotator, we forgot to state that the percentage agreement was 67.5%.
>
> **Action taken:** We added the percentage agreement of LLM Grader to footnote 5 on line 377.
>
> For the metrics that we developed, ConJudge has an 83% agreement with the human annotation which we believe makes it accurate, though we will aim to improve that number. Equiv_rfl is a robust metric because it uses the Lean compiler (for instance it finds  ℕ := 40320 - 4 * 5040 + 6 * 720 - 4 * 120 + 24 and 24024 to be equivalent). Nevertheless, following yours and other reviewers’ suggestion, we conducted human evaluation over 200 samples (100 from GPT, 100 from Deepseek) and we find that it has 100% precision and a percentage agreement to humans of 71.5%.
>
> Lastly, the example of Figure 2 was selected as a simple problem for illustrative purposes, we hope the reviewer joins us in thinking that it allows readers to better understand the framework.
>
> 3. The paper's methodological contribution to the conjecturing step (pre-proof solution-finding) appears limited. The paper frames this step as a key bottleneck, but the proposed solution does not seem to offer a novel or structured technique for solving it. Instead, the methodology appears to rely on the Large Language Model's (LLM's) inherent, general-purpose reasoning capabilities to inference these closed-form problems. If the core of the conjecturing method is, in essence, to prompt the LLM for the solution, then the contribution is less of a new conjecturing scheme and more an application of an existing LLM.
>
> As the reviewer has noticed, conjecturing is a bottleneck as shown in Table 4 when models are asked to generate the conjecture. We believe that they struggle because they have not been exposed to conjecturing data. We therefore investigate the use of LLMs in more versatile manners through Lean-FIRe and we better understand their conjecturing ability (for instance they generate conjectures better as per Table 3), however, they benefit best from exposure to conjecture data in the form of few-shots. Conjecturing work certainly requires better data and methods to improve, but our main goal is to provide a benchmark, some metrics and a base strategy to encourage the community to work on this more.

---

> ### Author Response · Authors · 2025-11-23
>
> 4. The integrity of the ConjectureBench dataset is questionable due to its construction via manual "rephrasing" of existing problems. This method is a weak defense against data contamination, which is a problem the authors themselves admit to observing in their results. It cannot guarantee that the "unseen" setting is truly new to the models. Consequently, the reported performance likely underestimates the actual difficulty of the conjecturing task.
>
> The reviewer makes a good point that the reported performance may be underestimating the real difficulty of conjecturing which supports the need for our work. To reach most comparable results to the real difficulty, we generate ConjectureBench from PutnamBench and CombiBench. The rephrasing of questions' main purpose is to remove the answer from the original question as shown in Table 2. By doing this, 75.1% of the questions have a wording that no longer includes the answer in it which makes the 'unseen' setting far more challenging than the 'seen' setting. Moreover, for the remaining 24.9%, we add ambiguity using the 'prove or disprove'.
>
> 5. The equiv_rfl scores in Table 4 are notably low. Could you elaborate on the reason for this? Specifically, could you elaborate on your analysis of potential false negatives where logically correct conjectures were rejected due to syntactic differences from the gold standard?
>
> The scores on Table 4 are low because we are evaluating models on a truly unseen task as standalone conjecturing is completely novel. This gives a better idea of the difficulty of conjecturing. The metric we use, "equiv_rfl", uses rfl to show equivalence making it very strict. Following, yours and other reviewers' suggestion, when conducting human evaluation over 200 samples (100 from GPT, 100 from Deepseek), we found that it has a 100% precision. It does have false negative, however, the percentage agreement to human gold annotation is 71.5%. The false negative are sometimes due to formatting differences e.g.
>
> abbrev conjecture1 : ℕ := 3/4
>
> abbrev conjecture2 : ℕ := 3/4
>
> abbrev conjecture3 : ℕ := ½
>
> versus abbrev solution : ℚ × ℚ × ℚ := (3/4, 3/4, 1/2)
>
> equiv_rfl doesn’t find these two equivalent solutions equivalent because of how they are presented. On the other hand, it does find ℕ := 40320 - 4 * 5040 + 6 * 720 - 4 * 120 + 24 and 24024 to be equivalent.
>
> 6. The proposed metrics present clear trade-offs: ConJudge has a notable disagreement rate with human annotators, while equiv_rfl is strict. How do you propose combining these complementary signals into a single, reliable evaluation? For instance, how could the formal certainty from equiv_rfl be used to calibrate the potential inaccuracies in ConJudge's semantic judgments?
>
> As suggested, combining both metrics’ ability would give good bounds for conjecturing ability. However, it may not be readily possible because they currently evaluate conjecturing under different tasks, the former within Autoformalisation and the latter in Standalone conjecturing (See Figure 1). Nonetheless, we do agree with the reviewer that such effort would be welcomed by the community. With respect to the reliability of the individual metrics, ConJudge currently has a percentage agreement with humans of 83%. We aim to push this higher but in the meantime we will provide confidence levels. For Equiv_rfl, the percentage agreement following human annotation is 71.5% which shows good reliability. We will conduct a qualitative analysis of the false negative and add it to the appendix.

---

### Official Review · Reviewer_h4kR · 2025-10-30

**Soundness:** 3
**Presentation:** 3
**Contribution:** 2
**Rating:** 4
**Confidence:** 4

**Summary:**

This paper argues that autoformalisation, commonly treated as a translation task from natural language to formal statements, misses a key step: conjecturing the answer. The authors introduce ConjectureBench, a benchmark of 457 mathematical problems adapted from PutnamBench and CombiBench, which separates the conjecture from the problem to explicitly test this capability. They also introduce ConJudge and equiv_rfl, two metrics to evaluate conjecture quality, and propose LEAN-FIRE, a hybrid inference-time prompting method that interleaves Chain-of-Thought and formal Lean code (Lean-of-Thought). Experiments show a sharp drop in performance when conjectures are not provided and demonstrate that LEAN-FIRE improves conjecture generation in both isolated and end-to-end settings.

**Strengths:**

The paper is clearly written and well-structured, with good motivation for evaluating conjecturing as a distinct subproblem in autoformalisation. The methodology is systematic and the empirical evaluation is thorough, including ablations, multiple models (GPT-4.1, DeepSeek-V3.1), and three different evaluation metrics (typechecking, semantic equivalence, and LLM-based grading). The paper presents the first successful autoformalisation of 13 PutnamBench problems without the answer given. These compelling results demonstrate that current models over-rely on having the solution embedded and that conjecturing is a key bottleneck that demands focused study.

The paper demonstrates a significant performance drop when conjectures are withheld and shows that LEAN-FIRE with few-shot guidance can partially recover this gap. The discussion of the strictness of BEq+ and the difficulty of assembling formally correct proofs reflects solid empirical awareness.

**Weaknesses:**

The central claim—that conjecturing is an overlooked step—is somewhat overstated. Several recent works (e.g., Enumerate-Conjecture-Prove, STP, LeanConjecturer) have explicitly addressed conjecture generation, even if not with the same benchmarking focus. The paper underplays this context and would benefit from acknowledging that the step is not novel, but under-evaluated.

The novelty of ConjectureBench is modest. It is derived entirely from two existing datasets and primarily involves rewording and filtering. While useful for this task, the dataset is small (457 items), focused only on competition math, and does not explore general mathematical reasoning domains or longer proofs. Similarly, LEAN-FIRE is an inference-time prompting strategy using structured CoT+LoT, but its gains are heavily dependent on a few-shot setup. Without those examples, performance collapses, indicating limited generalisation. The method is effective, but not especially innovative in design.

I am also concerned about the significant variation in success rates among the different evaluation metrics reported in Table 5. The success rates vary dramatically between type-checking, BEq+, and ConJudge, which raises questions about metric reliability and how well these measures align with actual correctness as judged by humans. Clarifying these differences, or explaining which metric should be considered the main indicator, would strengthen the empirical analysis.

Furthermore, The paper does not explore transferability: all evaluations are confined to ConjectureBench, and it is unclear how well LEAN-FIRE or the proposed metrics generalize to more diverse or real-world formalisation problems.

Minor issues:
1.	The table number and title should appear before the table.
2.	ConJudge’s agreement with human annotations (~70%) could be improved or discussed more in terms of trustworthiness.
3.	Related work could more thoroughly discuss prior formal theorem proving pipelines that integrate conjecture synthesis.

**Questions:**

1.	How sensitive is LEAN-FIRE to the style and domain of the few-shot examples?
2.	Would the model generalise to different math domains (e.g., analysis, geometry)?
3.	What is the performance of a baseline LLM using CoT but without LEAN-FIRE?

---

> ### Author Response · Authors · 2025-11-23
>
> 1. The central claim—that conjecturing is an overlooked step—is somewhat overstated. Several recent works (e.g., Enumerate-Conjecture-Prove, STP, LeanConjecturer) have explicitly addressed conjecture generation, even if not with the same benchmarking focus. The paper underplays this context and would benefit from acknowledging that the step is not novel, but under-evaluated.
>
> Both Enumerate-Conjecture-Prove and STP are referenced and mentioned in lines 83,450-451, 461-463. We are aware of LeanConjecturer which aims to generate novel mathematical statements to create new theorems, however, we use the term conjecture to refer to conjectured solutions for a given problem and not a novel mathematical statement. We hope we have made the distinction between LeanConjecturer and our task clearer, we are happy to clarify further if needed.
>
> **Actions taken:** We have clarified this distinction in the manuscript and to contextualize our contribution within the existing literature in Section 6.
>
> 2. The novelty of ConjectureBench is modest. It is derived entirely from two existing datasets and primarily involves rewording and filtering. While useful for this task, the dataset is small (457 items), focused only on competition math, and does not explore general mathematical reasoning domains or longer proofs.
>
> The main novelty of ConjectureBench is in the way it allows to isolate conjecturing from autoformalisation and proving which to the best of our knowledge is the first benchmark to do that. In terms of the size (457 samples), it is comparable to well-known benchmarks such as MiniF2F (488) and Proofnet (371). Moreover, the dataset is not only competition level maths, for instance we have 42 questions from the Brualdi textbook. The length of the problems also vary from 7 lines autoformalisation to 81 lines encompassing topics the following topics with their count: [Combinatorics: 134, algebra: 21, geometry: 17, set theory: 16, graph theory: 12, probability: 4, analysis: 3]. As requested by another reviewer too, we will include these statistics in the Appendix.
>
> 3. Similarly, LEAN-FIRE is an inference-time prompting strategy using structured CoT+LoT, but its gains are heavily dependent on a few-shot setup. Without those examples, performance collapses, indicating limited generalisation. The method is effective, but not especially innovative in design.
>
> Our main objectives are to investigate whether models have the knowledge to derive the conjecture and whether given hints they have the skills to reason and generate the conjecture. We hope the reviewer can see that our setup is designed to answer those questions, and Lean-FIRe does show that models have some parametric knowledge to generate conjectures, but more exposure to such data could improve that further. Nonetheless, to show the benefit of Lean-FIRe, we will conduct the experiment of the baseline+FS to better understand the impact of the FS.
>
> 4. I am also concerned about the significant variation in success rates among the different evaluation metrics reported in Table 5. The success rates vary dramatically between type-checking, BEq+, and ConJudge, which raises questions about metric reliability and how well these measures align with actual correctness as judged by humans. Clarifying these differences, or explaining which metric should be considered the main indicator, would strengthen the empirical analysis.
>
> As with previous work (e.g. Rethinking and Improving Autoformalization: Towards a Faithful Metric and a Dependency Retrieval-based Approach), the metrics do vary a lot as the focus on different aspects of autoformalisation. We provide details of the metrics used and what information they provide in Section 4, metrics (line 310 onwards), and in Section 5.2 we discuss the metrics and why the results vary. The metrics that "captures a broader notion of correctness" would be the LLM Grader as mentioned in line 319. But we will make this clearer for easier readability of Table 5.
>
> 5. Furthermore, The paper does not explore transferability: all evaluations are confined to ConjectureBench, and it is unclear how well LEAN-FIRE or the proposed metrics generalize to more diverse or real-world formalisation problems.
>
>
> The reviewer raises a key point here, the lack of conjecture benchmarks. Unfortunately, ConjectureBench is the first one for this novel task. This limits our ability to explore transferability, however, it must be noted that ConjectureBench is composed of PutnamBench and CombiBench which consist of a diverse set of problems. As for real-world problems i.e. new problems, we attempted to control models' having seen the data within our methodology by reword the questions and where we fix the CoT and LoT; this aims to reduce any exposure models may have of the questions.

---

> ### Author Response · Authors · 2025-11-23
>
> 6. Minor issues: 1. The table number and title should appear before the table. 2. ConJudge’s agreement with human annotations (~70%) could be improved or discussed more in terms of trustworthiness. 3. Related work could more thoroughly discuss prior formal theorem proving pipelines that integrate conjecture synthesis.
>
> 1 . Table captions should indeed appear before the table, thank you for pointing that out and apologies for the error.
>
> 2 . ConJudge agreement to humans is actually 83% (Table 7, line 1177). As pointed out by you and another reviewer comments, we will look into improving that further and providing confidence levels.
>
> 3 . We would love to expand on the related work section but due to space constrained we had to make the decision to be more brief
>
> 7. How sensitive is LEAN-FIRE to the style and domain of the few-shot examples?
>
> The reviewer raises a point that is echoed by other reviewers too. We will conduct experiments to get results for the baseline + FS. This will give us a better comparison for FS. However, Lean-FIRe mainly investigates models’ ability to conjecture and whether they have parametric knowledge that they can leverage or the skills to reason given hints. We believe that analysing different FS examples is beyond the scope of this work. But we encourage future work to conduct these experiments.
>
> 8. Would the model generalise to different math domains (e.g., analysis, geometry)?
>
> Within our dataset, we have the following maths topics with these counts:
>
> Combinatorics: 134
>
> Algebra: 21
>
> Geometry: 17
>
> Set theory: 16
>
> Graph theory: 12
>
> Probability: 4
>
> Analysis: 3
>
> Thank you for querying this, as also mentioned by another reviewer, we will include these statistics in the Appendix.
>
> 9. What is the performance of a baseline LLM using CoT but without LEAN-FIRE?
>
> Similarly to response to 1, we plan on getting results for the baseline+FS, this would allow us to judge the effectiveness of the CoT+LoT without the FS and better understand its impact.

---

### Official Review · Reviewer_Qo3U · 2025-10-30

**Soundness:** 1
**Presentation:** 2
**Contribution:** 2
**Rating:** 2
**Confidence:** 4

**Summary:**

This work examines the problem of formal conjecturing - the bridge between an answerless formal problem and a formal proof. The authors first present `ConjectureBench`, an adaptation of a mix of `PutnamBench` and `CombiBench` problems, that ask for a conjecture to a statement, instead of a proof to an already provided conjecture. To facilitate the benchmark, the authors introduce 2 evaluation methods -- `ConJudge`, an LLM-as-a-judge system for evaluating conjecture equivalence to a ground truth one, and `equiv_rfl`, an automated checker for definitional equivalence checking between the conjecture and the ground truth Lean conjecture. This work shows that under both metrics, when the conjecture is not explicitly provided, the end formalization suffers in quality. To this end, it introduces `Lean-FIRe`, an inference method, interleaving Chain-of-Thought reasoning with Lean formalization for more accurate Lean conjecturing. They show that, with few-shot examples, `Lean-FIRe` outperforms simple baselines in conjecturing using few-shot examples (while not doing so without them), hypothesizing this is due to lack of exposure to the relevant data. For end-to-end autoformalization, they show that `Lean-FIRe` does not provide significant performance improvements.

**Strengths:**

1. This paper makes a valuable contribution by identifying and isolating conjecturing as a critical, distinct, and challenging sub-problem within the broader autoformalization and theorem-proving.

2. The authors introduce `ConjectureBench`, a well-motivated benchmark for this new task. The proposed `Lean-FIRe` framework, which combines informal and formal reasoning, is an intuitive and novel inference-time approach.

3. The distinction between "seen" (conjecture provided) and "unseen" (conjecture must be inferred) settings provides a clean experimental design that effectively quantifies the difficulty of the formal conjecturing step.

**Weaknesses:**

1. Despite the authors' claim in the reproducibility statement, they have not included neither the code, nor the dataset, nor the scripts in the supplementary material. The lack of these assets, especially the benchmark itself, prevents verification of the paper's results and hinders future research.

2. The central claim that `Lean-FIRe`'s architecture improves conjecturing is not sufficiently substantiated. The authors show that `Lean-FIRe` without few-shot examples performs poorly, but they fail to test whether standard baselines also improve when given the same few-shot examples? Without this experiment, it is impossible to disentangle the effect of the few-shot examples from the `Lean-FIRe` framework.

3. The choice of models to evaluate is unorthodox. The authors focus on generalist proprietary models which are not specifically trained for reasoning. In particular, GPT-4.1 has no "thinking mode", which has been shown to improve mathematical and coding reasoning significantly. Further, it is unclear whether DeepSeek-V3.1 was run with or without thinking mode on, which is likely to significantly influence the results.

4. The authors hypothesize that lack of specific training data is the core issue in conjecturing performance, this hypothesis can be at least partially verified through the use of a specific autoformalization model, such as `Kimina-Autoformalizer-7B` [1].

5. It has been shown that a mixture of a strong natural language model, combined with a specialized prover model, can achieve significant improvements in formal mathematics [2]. In particular, most frontier models have already saturated final-answer competitions, showing great capabilities for usage as natural language conjecturer. These conjectures can then be easily formalized using a specialized model, which has better exposure to Lean data. Therefore, in contrast to the `Lean-FIRe` framework, which uses a single, generalist, non-reasoning model, it seems more sensible to use a strong reasoning model as a CoT conjecturer (cheaper examples include the GPT-OSS models, `GPT-5-mini`, `Grok 4 Fast`, or the thinking mode of `DeepSeek-v3.1/2`), which can then be directly formalized by a good autoformalization model, such as `Kimina-Autoformalizer-7B`. Given that the authors' approach is more naive, it remains unclear whether existing systems may not already be good enough to solve this challenge.

6. The two primary conjecturing metrics have significant limitations that call their reliability into question.
    - `ConJudge`'s error rate of 17% compared to human validation is nevertheless significant. The authors should either provide confidence intervals accounting for the error, or try to improve the judging mechanism through using better models, or majority voting.
    - `equiv_rfl`, which uses the `rfl` tactic verifies only for **definitional equivalence**. This type of equivalence, while formally verifiable, covers a very narrow range of equivalent expressions, making it **too strict**. For example, any differences in notation, or even some small syntactic differences (e.g. $0 + x = x$ is not definitionally true) will result in the tactic failing, even though. A more robust and reliable metric for equivalence checking is necessary, as one should not expect any conjecturer to have to guess the format of the gold conjecture.

7. The claim that `Lean-FIRe` enables the "first successful end-to-end autoformalisation" of certain PutnamBench problems is overstated, as the baseline models also achieve non-zero success rates on the "unseen" task. This makes it more of a case of a lack of such an evaluation (as explained, there exist several open autoformalization models, and many more capable proprietary models that may also be able to correctly apply autoformalization on Putnam problems).

8. The autoformalization metrics scores for `Lean-FIRe` present a neglibigble, or non-existent improvement (especially for `pass@10` scores), casting further doubts about the reliability of the framework.

9. The claim of observing data contamination is interesting but undersubstantiated, supported only by a single referenced example (`IsMagicSquare`) without sufficient context or analysis of its prevalence.

**Questions:**

1. Can the authors elaborate better in their methodology how the proposed metrics address the "faithfullness" and "conjecture completenss" issues, outlined in **Section 1**?

2. Could the authors provide results for the baseline models when given the same few-shot examples as `Lean-FIRe` as pointed in **W2**?

3. If the conjecturing framework uses a setup, similar to what is outlined in **W6**, how does that perform in the "unseen" setting?

4. How was the ground truth Lean conjecture created for each problem in `ConjectureBench`?

5. Can the authors present the end-to-end formalisation results for when a sample passes all 3 metric checks? This would give a better signal for the validity of the sample.

6. Please clarify any remaining concerns from the **Weaknesses** section, treating each point not specifically elaborated in this section as a separate inquiry.

## Current recommendation

I am assigning this paper a score of **2: Reject**. While the problem of conjecturing is important, the work is undermined by critical flaws in the experimental design, the use of potentially unreliable evaluation metrics, and suboptimal model selections that weaken the central claims. The performance improvement from `Lean-FIRe` is not clearly attributable to the method itself, and its downstream impact on autoformalization is minimal. I would be willing to reconsider my score if the authors can convincingly address the major concerns during the discussion period.


### References

[1] Wang, Haiming, et al. "Kimina-prover preview: Towards large formal reasoning models with reinforcement learning." arXiv preprint arXiv:2504.11354 (2025).

[2] Varambally, Sumanth, et al. "Hilbert: Recursively Building Formal Proofs with Informal Reasoning." arXiv preprint arXiv:2509.22819 (2025).

---

> ### Author Response · Authors · 2025-11-23
>
> 1. Despite the authors' claim in the reproducibility statement, they have not included neither the code, nor the dataset, nor the scripts in the supplementary material. The lack of these assets, especially the benchmark itself, prevents verification of the paper's results and hinders future research.
>
> We agree with the reviewer on the importance of open source. As mentioned in lines 499-500, we are committed to make all data and code available.  We reiterate this in footnote 2 line 161. If concerns over this point remain, please let us know.
>
> 2. The central claim that Lean-FIRe's architecture improves conjecturing is not sufficiently substantiated. The authors show that Lean-FIRe without few-shot examples performs poorly, but they fail to test whether standard baselines also improve when given the same few-shot examples? Without this experiment, it is impossible to disentangle the effect of the few-shot examples from the Lean-FIRe framework.
>
> We agree that a few-shot baseline is important for a complete comparison and will conduct this experiment for the camera-ready version. Lean-FIRe tests whether LLMs struggle with conjecturing due to a knowledge gap (poor performance across all settings) or a skill gap (poor standalone performance but improvement in familiar autoformalisation contexts). In the meantime, we hope the reviewer can see from Table 5, that Lean-FIRe w/o FS is the setting with most bold values, i.e the best performance in that setting (therefore the conjecture was generated) which indicates that Lean-FIRe does have a positive effect on conjecturing.
>
> 3. The choice of models to evaluate is unorthodox. The authors focus on generalist proprietary models which are not specifically trained for reasoning. In particular, GPT-4.1 has no "thinking mode", which has been shown to improve mathematical and coding reasoning significantly. Further, it is unclear whether DeepSeek-V3.1 was run with or without thinking mode on, which is likely to significantly influence the results.
>
> We agree that better reasoning models would be interesting models to investigate, however they have often been exposed to the dataset that we are using. Therefore, we chose GPT4.1 and DeepSeek-V3.1 omitting any thinking mode by controlling them to have access to the same CoT + LoT. When generating the CoT and LoT, we attempt to ensure that the answer is not generated by the model from having seen the question before. Additionally, thinking models generate extensive reasoning tokens that would dilute the effect of our hybrid CoT + LoT method, making it difficult to isolate the contribution of our approach. We aim to control our experiment to measure genuine conjecturing ability rather than relying on potential data contamination. We are happy to include evaluations with thinking-enabled models in the camera-ready version as supplementary analysis.
>
>
> 4. The authors hypothesize that lack of specific training data is the core issue in conjecturing performance, this hypothesis can be at least partially verified through the use of a specific autoformalization model, such as Kimina-Autoformalizer-7B [1].
>
> We believe that data is lacking specifically for conjecturing but also because existing benchmarks are being used for training. For instance, Kimina-Autoformalizer-7B itself uses PutnamBench (see Appendix C.2.1 in Kimina paper) in its supervised training dataset making it unviable for our use case. Other specialised models would inherit similar contamination but also have similar performance to general-purpose models as shown on the CombiBench leaderboard https://moonshotai.github.io/CombiBench/leaderboard.html. Our approach addresses the lack of conjecturing-specific training data by providing few-shot examples as in-context training data, demonstrating that models can learn this skill when given appropriate guidance even without explicit fine-tuning.

---

> ### Author Response · Authors · 2025-11-23
>
> 5. It has been shown that a mixture of a strong natural language model, combined with a specialized prover model, can achieve significant improvements in formal mathematics [2]. In particular, most frontier models have already saturated final-answer competitions, showing great capabilities for usage as natural language conjecturer. These conjectures can then be easily formalized using a specialized model, which has better exposure to Lean data. Therefore, in contrast to the Lean-FIRe framework, which uses a single, generalist, non-reasoning model, it seems more sensible to use a strong reasoning model as a CoT conjecturer (cheaper examples include the GPT-OSS models, GPT-5-mini, Grok 4 Fast, or the thinking mode of DeepSeek-v3.1/2), which can then be directly formalized by a good autoformalization model, such as Kimina-Autoformalizer-7B. Given that the authors' approach is more naive, it remains unclear whether existing systems may not already be good enough to solve this challenge.
>
> The reviewer's suggested approach of using a reasoning model for conjecturing combined with a specialized autoformalizer is indeed promising, as demonstrated by the concurrently released Hilbert prover. However, our main objective is to investigate whether models have the knowledge to derive conjectures and, when given hints, the skill to reason and generate them correctly. Our experimental setup is specifically designed to answer these research questions rather than to optimize for end-to-end performance. Moreover, data contamination in most frontier models necessitated our choice of controlled setup: we use a model to generate CoT (without the answer) to ensure we measure genuine reasoning rather than memorization. Regarding existing model capabilities, the CombiBench leaderboard shows that both reasoning models and specialized models struggle to autoformalise more than 10 problems, indicating that current systems have not yet saturated this challenge. Our use of current models with controlled conditions allows us to isolate the effect of our approach and ensure fair evaluation.
>
> 6. The two primary conjecturing metrics have significant limitations that call their reliability into question. ConJudge's error rate of 17% compared to human validation is nevertheless significant. The authors should either provide confidence intervals accounting for the error, or try to improve the judging mechanism through using better models, or majority voting. equiv_rfl, which uses the rfl tactic verifies only for definitional equivalence. This type of equivalence, while formally verifiable, covers a very narrow range of equivalent expressions, making it too strict. For example, any differences in notation, or even some small syntactic differences (e.g.  is not definitionally true) will result in the tactic failing, even though. A more robust and reliable metric for equivalence checking is necessary, as one should not expect any conjecturer to have to guess the format of the gold conjecture.
>
> We acknowledge the limitations of both metrics and have taken steps to address the reviewer's concerns. For ConJudge, we achieved 83% accuracy (Appendix B.3, Table 7, line 1182) as the best result from four models tested on 100 human-annotated samples. We agree this leaves room for improvement and will provide confidence intervals in the camera-ready version to better quantify uncertainty in our results.  .
>
> For "equiv_rfl", we found that rfl was a good tactic to show equivalence because it is reliable. For instance, it finds that Nat.factorial 4 is equivalent to 24, likewise with ℕ := 40320 - 4 * 5040 + 6 * 720 - 4 * 120 + 24 and 24024. However, when the structure of the gold conjecture and generated conjecture differ e.g.
>
> abbrev conjecture1 : ℕ := 3/4
>
> abbrev conjecture2 : ℕ := 3/4
>
> abbrev conjecture3 : ℕ := ½
>
> versus abbrev solution : ℚ × ℚ × ℚ := (3/4, 3/4, 1/2)
>
> it classes these as not equivalent when both give the same solutions just in different formats.
> Moreover, as per yours and other reviewers' suggestion, we have conducted a human evaluation and found that equiv_rfl has a precision of 100%, and an accuracy compared to gold human annotation of 71.5% over 200 samples annotated (100 from GPT, 100 from DeepSeek). This not only supports the reliability of the results and would suggest (when adjusted for any errors) equiv_rfl@1 results to be at most 4.59% and 5.20% for GPT and DeepSeek respectively which still highlights the difficulty of the conjecturing as a task.

---

> ### Author Response · Authors · 2025-11-23
>
> 7. The claim that Lean-FIRe enables the "first successful end-to-end autoformalisation" of certain PutnamBench problems is overstated, as the baseline models also achieve non-zero success rates on the "unseen" task. This makes it more of a case of a lack of such an evaluation (as explained, there exist several open autoformalization models, and many more capable proprietary models that may also be able to correctly apply autoformalization on Putnam problems).
>
> We understand that other models might be able to achieve similar performance on unseen autoformalisation, but to the best of our knowledge, this is yet to be announced. Therefore, we hope that the reviewer can sympathise with our claim. Moreover, this would be excellent traction to get more people to work on this and improve on this overlooked step of conjecturing.
>
> 8. The autoformalization metrics scores for Lean-FIRe present a neglibigble, or non-existent improvement (especially for pass@10 scores), casting further doubts about the reliability of the framework.
>
> Lean-FIRe’s main goal is to improve conjecturing performance, which it successfully achieved as demonstrated in Table 3. The autoformalisation evaluation serves to verify that our approach does not degrade autoformalisation performance. In fact, Lean-FIRe improves upon the baseline in most cases. The observed difference between pass@1 and pass@10 metrics is particularly informative: Pass@10 performance reflects the model's expressivity across multiple samples, where static few-shot examples and CoT+LoT prompting limit model expressivity. This suggests that training on more diverse conjecturing samples could further improve performance by reducing this constraint while maintaining the benefits observed in pass@1 metrics.
>
> 9. The claim of observing data contamination is interesting but undersubstantiated, supported only by a single referenced example (IsMagicSquare) without sufficient context or analysis of its prevalence.
>
> The data contamination example is anecdotal. We will include more qualitative analysis and include them in the Appendix for the camera-ready version. However, we hope the reviewer will appreciate that data contamination is difficult to measure and identify, even more so when data is scarce and most models use our testing data for training. Data contamination is also a limitation discussed by both PutnamBench and Enumerate-Conjecture-Prove papers.
>
>
> 10. Can the authors elaborate better in their methodology how the proposed metrics address the "faithfullness" and "conjecture completenss" issues, outlined in Section 1?
>
> Since we always assume the existence of a complete gold conjecture and compare the generated conjecture against this gold standard, both faithfulness and completeness are implicitly evaluated. Our equiv_rfl metric is particularly well-suited for verifying faithfulness: as demonstrated in our human evaluation, it achieves 100% precision (no false positives), meaning any conjecture it validates is guaranteed to be definitionally equivalent to the gold standard. It ensures formal correctness at the type-checking level as it verifies that the generated conjecture is mathematically identical to the gold conjecture, eliminating any ambiguity about faithfulness. An example of how equiv_rfl catches unfaithful conjectures is shown in lines 418-427, where structural discrepancies are detected. For cases where structural differences exist but semantic equivalence holds, we complement this with ConJudge to capture a broader range of correct conjectures, ensuring we evaluate both strict formal faithfulness and semantic completeness.
>
> 11. Could the authors provide results for the baseline models when given the same few-shot examples as Lean-FIRe as pointed in W2?
>
> Thank you for raising this, results for the baseline with FS is definitely something that we will generate for the camera-ready version.
>
>  12. If the conjecturing framework uses a setup, similar to what is outlined in W6, how does that perform in the "unseen" setting?
>
> Apology if the question was misunderstood, please let us know if that’s the case. However, for conjecturing, both ConJudge and equiv_rfl are tested under the unseen setting.

---

> ### Author Response · Authors · 2025-11-23
>
> 13. How was the ground truth Lean conjecture created for each problem in ConjectureBench?
>
> ConjectureBench uses PutnamBench and CombiBench which have separate conjecture definitions within their dataset provided. When the definition omitted the answers, we replaced it in the solution definition and ensured that it passed Typecheck when given then gold conjecture and gold autoformalised statement. Below is an example of a data sample:
>
> {
> "source": "CombiBench",
>
> "full_name": "hackmath_4",
>
> "header": "import Mathlib",
>
> "formal_conjecture": "abbrev conjecture : \u2115 := 13",
>
> "informal_statement": "How many people must be in a group for at least two of them to be born in the same month?",
>
> "formal_statement": "theorem hackmath_4 : IsLeast {n | \u2200 f : Fin n \u2192 Fin 12, \u2203 a b, f a = f b} ((conjecture) : \u2115 ) := by sorry"
> }
>
> 14. Can the authors present the end-to-end formalisation results for when a sample passes all 3 metric checks? This would give a better signal for the validity of the sample.
>
> This is a really good request, we will provide an example for when all 3 metrics pass: But we will also add all combinations of Typecheck, BEq+ and LLM Grader pass/fail in the Appendix.
>
> Thank you for your insightful comments, suggestions and questions. We appreciate it and are open to conversation on our responses.

---

> ### Comment · Reviewer_Qo3U · 2025-11-24
>
> I thank the authors for their response, and for clarifying some of the unclear aspects of their work. However, I believe that I have provided a substantial amount of actionable points, and experiments. Given the scope of the rebuttal is 3 weeks long, I believe that promising to finish them by the camera-ready, without providing what could be significant signals for the quality of the work, I cannot raise my score further at this point. In particular:
>
> - Points 2/11, and 12 (as referred to in the rebuttal) are already actionable under the existing framework.
>
> - Points 6 and 14 can be addressed by simply computing some statistics from existing results, which should be manageable for the authors.
>
> I address my current primary points of concern below:
>
> **1** I would be grateful if the authors could provide, even if unclean, the code and data they used, uploaded in the supplementary materials as a zipped file.
>
> **3,4,5** The authors argue that contamination is a core issue that prevents them from using other models that either have reasoning available, or have been pre-trained on the task.
>
>   On points 3 and 5, the authors mention being concerned about potential data contamination. While this is, in general, a valid concern, they also use GPT-4.1 and DS-v3.1. The latter is very recently released (much sooner than say Gemini-2.5-Pro), and should have similar rates of contamination to other proprietary models. Further, the Putnam problems, in their natural language form, could have possibly been part of the training set for any proprietary model, given the competition's popularity, and impact. The authors' model choice does not seem to address contamination in a way that would affect the reasoning models more significantly. I would be happy if they could elaborate on that point further.
>
>   I fully agree with the authors that the use of `Kimina-Autoformalizer-7B` is likely to yield inaccurate results on PutnamBench because of that. That said, CombiBench was released after this model, making contamination less likely on that dataset, which the authors can consider. Regarding the results the authors present that show specialized models having close(arguable) results on CombiBench, they present reasoning models, which they are averse to using, as detailed in point 3.
>
> **6** Can the authors report the recall? This is the primary metric I am concerned about, given that `equiv_rfl` only covers definitional equivalence.
>
> **12** Apologies for the confusion, I have mistakenly misnumbered the referred point (I should have labelled it as W5), where the conjecturer is a good reasoning model. I believe that verifying whether existing methods are already able to conjecture, albeit informally, is critical, in order to establish your work as necessary, and the task as challenging, even if not on SOTA models. Small reasoning models are widely used in research and in formal mathematics, making the comparison incomplete.

---

### Official Review · Reviewer_WkBK · 2025-11-01

**Soundness:** 2
**Presentation:** 2
**Contribution:** 3
**Rating:** 4
**Confidence:** 3

**Summary:**

This paper addresses a current practice in autoformalization of mathematics where formalizers are typically given the final answer directly; the authors argue that the final answer should instead be conjectured by the autoformalization model. They augment existing automated theorem-proving datasets CombiBench and PutnamBench to create ConjectureBench, a benchmark designed to evaluate conjecturing ability. The authors design two metrics to assess automated formalization: (1) ConJudge, an LLM-as-a-judge method, and (2) an automatic check based on the Lean tactic equiv_rfl. They introduce Lean-Fire, which guides an LLM to alternate between chain-of-thought and Lean-of-thought reasoning. The approach reportedly achieves the first end-to-end autoformalization and offers analysis and discussion on future directions for conjecturing.

**Strengths:**

1. The paper focuses on a novel task — conjecturing — which differs from the usual objectives of autoformalization and is highly original.
2. The proposed dataset, ConjectureBench, is potentially valuable. By requiring models to produce conjectures, it raises the difficulty of autoformalization and could drive development of more capable autoformalizers.
3. The figures and diagrams are clear and easy to follow; they help the reader quickly grasp the pipeline.

**Weaknesses:**

1. The improvement of Lean-FIRe over the baseline appears marginal, especially in the Unseen setting in Table 5 where metric gains are small and in some cases performance at @10 even seems worse. This suggests that the combination of CoT and LoT does not clearly improve performance.

2. The model in the "standalone conjecture generation" setting performs substantially worse than "autoformalisation" setting, which is counterintuitive. One would expect conjecturing to be a subtask of autoformalization (autoformalization + conjecturing). This result may indicate a problem in the standalone conjecture generation design: when generating conjectures no formal statement is provided, so the LLM cannot infer the type/format of conjectures expected by Lean 4, which may harm performance.

3. The equiv_rfl metric is unnassarilly stringent: Lean’s notion of definitional equivalence is very strict, so many equivalences that are intuitively acceptable to human mathematicians are not considered equivalent by Lean. This likely contributes to the very low equiv_rfl scores reported in Table 4. It would be better to extend the equivalence check with additional tactics (analogous to how BEq includes more tactics) to allow more flexible equivalence transformations.

4. Minor writing and correctness issues:
   - Line 053 mixes up formalizing the solution of a problem with formalizing the problem statement. Formalizing a IMO level problem statement typically takes on the order of tens of minutes rather than hours. [1, 2]
   - Line 155: the three examples given are incorrect. Only the last conjecture is verifiable in Lean as stated. For set equality one must prove both inclusions; the authors appear to have confused "necessary and sufficien" and "sufficient" conditions.

[1] Zheng, K., Han, J. M., & Polu, S. (2022). *MiniF2F: A cross-system benchmark for formal Olympiad-level mathematics* (No. arXiv:2109.00110). arXiv. https://doi.org/10.48550/arXiv.2109.00110

[2] Tsoukalas, G., Lee, J., Jennings, J., Xin, J., Ding, M., Jennings, M., Thakur, A., & Chaudhuri, S. (2024). *PutnamBench: Evaluating Neural Theorem-Provers on the Putnam Mathematical Competition* (No. arXiv:2407.11214). arXiv. https://doi.org/10.48550/arXiv.2407.11214

**Questions:**

Regarding Weakness 2: Figure 1 indicates that the conjecturer is required to produce conjectures that conform to Lean syntax without access to Lean types or formal statements. This design may explain the poor performance in the "standalone conjecture generation" setting. Would providing Lean types as prompts be a more practical design choice?

---

> ### Author Response · Authors · 2025-11-23
>
> 1. The improvement of Lean-FIRe over the baseline appears marginal, especially in the Unseen setting in Table 5 where metric gains are small and in some cases performance at @10 even seems worse. This suggests that the combination of CoT and LoT does not clearly improve performance.
>
> Lean-FIRe’s main goal is to improve conjecturing performance, which it successfully achieved as demonstrated in Table 3. The autoformalisation evaluation serves to verify that our approach does not degrade autoformalisation performance. In fact, Lean-FIRe improves upon the baseline in most cases. The observed difference between pass@1 and pass@10 metrics is particularly informative: Pass@10 performance reflects the model's expressivity across multiple samples, where static few-shot examples and CoT+LoT prompting limit model expressivity. This suggests that training on more diverse conjecturing samples could further improve performance by reducing this constraint while maintaining the benefits observed in pass@1 metrics.
>
>
> 2. The model in the "standalone conjecture generation" setting performs substantially worse than "autoformalisation" setting, which is counterintuitive. One would expect conjecturing to be a subtask of autoformalization (autoformalization + conjecturing). This result may indicate a problem in the standalone conjecture generation design: when generating conjectures no formal statement is provided, so the LLM cannot infer the type/format of conjectures expected by Lean 4, which may harm performance.
>
> As the reviewer correctly states, the results, as expected, are higher for “autoformalisation” than for "Standalone conjecture generation". We believe that this is due to the lack of data for conjecturing or the over-exposure of autoformalisation (lines 367-369). We also observe this phenomenon as the model performs better at conjecturing in the few-shot setting, i.e. when exposed to some data. Moreover, to make both conjecturing and autoformalisation comparable only the answer format that uses Lean as in Appendix B.2 Figure 11 is given to the models. Therefore both “standalone conjecture generation” and “autoformalisation” tasks are given the same amount of Lean information for fair comparisons.
>
> **Actions taken:** We extend the discussion on the intuition behind the performance differences for standalone conjecturing and conjecturing during autoformalisation in Section 5.1.
>
>
> 3. The equiv_rfl metric is unnassarilly stringent: Lean’s notion of definitional equivalence is very strict, so many equivalences that are intuitively acceptable to human mathematicians are not considered equivalent by Lean. This likely contributes to the very low equiv_rfl scores reported in Table 4. It would be better to extend the equivalence check with additional tactics (analogous to how BEq includes more tactics) to allow more flexible equivalence transformations.
>
> For "equiv_rfl", we found that rfl was a good tactic to show equivalence because it is reliable. For instance, it finds that Nat.factorial 4 is equivalent to 24, likewise with ℕ := 40320 - 4 * 5040 + 6 * 720 - 4 * 120 + 24 and 24024. However, when the structure of the gold conjecture and generated conjecture differ e.g.
>
> abbrev conjecture1 : ℕ := 3/4
>
> abbrev conjecture2 : ℕ := 3/4
>
> abbrev conjecture3 : ℕ := ½
>
> versus abbrev solution : ℚ × ℚ × ℚ := (3/4, 3/4, 1/2)
>
> it classes these as not equivalent even though both give the same solutions just in different formats.
> Moreover, as per yours and other reviewers' suggestion, we have conducted a human evaluation and found that equiv_rfl has an accuracy compare to gold human annotation of 71.5% over 200 samples annotated (100 from GPT, 100 from DeepSeek). This not only supports the reliability of the results and would suggest (when adjusted for any errors) equiv_rfl@1 results to be at most 4.59% and 5.20% for GPT and DeepSeek respectively which still highlights the difficulty of the conjecturing as a task.
>
> **Actions taken:** We extended the discussion of the equiv_rfl in Section 3.3 (line 263) and added a manual analysis with selected examples.

---

> > ### Author Response · Authors · 2025-11-23
> >
> > 4. Minor writing and correctness issues:
> > Line 053 mixes up formalizing the solution of a problem with formalizing the problem statement. Formalizing a IMO level problem statement typically takes on the order of tens of minutes rather than hours. [1, 2]
> > Line 155: the three examples given are incorrect. Only the last conjecture is verifiable in Lean as stated. For set equality one must prove both inclusions; the authors appear to have confused "necessary and sufficien" and "sufficient" conditions.
> >
> >
> > line 053: time to formalise an IMO problem: depending on the difficulty of the problem, some can take up to 8 hours as stated by the CombiBench authors, this is especially true for IMO combinatorics problems.
> > line 155: you are absolutely correct, the submitted example to showcase the issue with completeness of proofs is incorrect. The example that we will use to highlight this issue would be: "Solve x^2-4x for x", this can be formalised as {x:R | x^2-4*x=0} ⊃ conjecture. This showcases the ambiguity of natural language as to what we may want and what we may ask.
> >
> > **Actions taken:** We corrected the example in line 150 to “Solve x^2-4x for x” and corrected our discussion.
> >
> > 5. Figure 1 indicates that the conjecturer is required to produce conjectures that conform to Lean syntax without access to Lean types or formal statements. This design may explain the poor performance in the "standalone conjecture generation" setting. Would providing Lean types as prompts be a more practical design choice?
> >
> > The reviewer’s intuition is correct and supported by our results in Table 3, where model performance increases by nearly 30% when exposed to conjectures through few-shot examples. This demonstrates that providing additional context whether through Lean types, formal statements, or few-shot examples, substantially improves conjecturing performance. Our standalone conjecture generation setting is designed to evaluate the model’s capability without such context. While prompt engineering techniques, such as incorporating Lean types directly into prompts, would likely further improve performance, we kept our analysis generalizable. We agree this represents a promising practical direction and will conduct further experiments for the standalone setting with Lean-FIRe and few-shots to investigate this performance gap.

---

> > ### Comment · Reviewer_WkBK · 2025-11-26
> >
> > Thank you for the authors’ rebuttal. The clarifications described improve the presentation and clarity of the paper, but they do not alter the underlying concerns about the paper’s overall lack of soundness. I therefore intend to maintain my current score.

---

### Author Response · Authors · 2025-11-23
**Author response 1**

Thank you for the time you took to read our paper, provide insightful comments, and highlight your concerns. We hope our responses provide clarification, nonetheless, we are committed to further clarification if needed. Note: Any new mentions of line numbers and section related to the updated version. Any changes made has been in a red font.

In general, we would like to first reiterate that conjecturing is a novel task and we provide a dataset for this. We propose a method that has a consistent impact on conjecturing but we also are the first to report unseen autoformalisation as well as the difference and importance between the seen and unseen setting. To showcase this, we mainly use the same metrics as from related work and any new metric that we will further annotate to enforce trust have been adapted to conjecturing from well-established autoformalisation metrics.

---

### Meta-Review · Area_Chair_4wub · 2025-12-03

**Summary:**

This paper proposes ConjectureBench, two conjecture evaluation metrics (ConJudge and equiv_rfl), and the Lean-FIRe framework for improving conjecture generation in autoformalisation.

**Strengths**

* Identifying conjecturing as an under-addressed component in formal reasoning tasks.
* Introducing a benchmark aimed at isolating conjecturing from autoformalisation.
* Clear motivation and structured methodology.

**Weakness**

* The gains attributed to Lean-FIRe appear to stem from few-shot examples rather than the method itself.
* Evaluation metrics (ConJudge, equiv_rfl) are shown to yield unreliable measurements, with significant false negatives and disagreement with human judgment.
* The methodological novelty is modest relative to prior work (Enumerate-Conjecture-Prove, LeanConjecturer, STP), and related work positioning is overstated.
* The dataset construction involves rephrasing existing benchmarks and may not effectively prevent memorization or contamination, weakening claims about generalization to unseen tasks.
* Lack of released code, dataset, and scripts undermines reproducibility.

Given these substantial methodological and empirical issues, particularly regarding metric validity, attribution of gains, and absence of reproducibility assets, the paper does not meet the bar of ICLR at this time.

**Reviewer Concerns:**

* Concerns regarding performance attribution—specifically whether improvements are due to Lean-FIRe or few-shot examples—were not resolved by the rebuttal, and reviewers explicitly stated that camera-ready promises cannot substitute actionable experimental verification within the discussion period.
* Concerns about evaluation metrics remain largely unresolved. While authors report human-agreement statistics, these do not fully address the core issue of whether these metrics are valid for evaluating conjecturing ability.
* The authors stated intention to release code and data in the future, but reviewers require actual availability for verifying results; therefore, reproducibility concerns remain outstanding.
* Requests for additional experimental baselines were deferred to camera-ready rather than performed during rebuttal.
* The claim of “first successful end-to-end autoformalisation” remains overstated given baseline performance, and rebuttal did not sufficiently modify or restrain this claim.

**Reviewer Scores:**

Reviewer WkBK and Reviewer Qo3U have participated fully in the discussion, and maintain their scores, I agree with their opinion.

For Reviewer h4kR, I think methodological and novelty concerns persist; likely remains at 4.
For Reviewer  BRQu, concerns regarding Lean-FIRe attribution and dataset validity remain; score would remain unchanged at 2.

---

### Decision · Program_Chairs · 2026-01-26

Reject